# Why Do Pretrained Language Models Help in Downstream Tasks? An Analysis of Head and Prompt Tuning

**Colin Wei**     **Sang Michael Xie**     **Tengyu Ma**

Department of Computer Science
Stanford University

{colinwei,xie,tengyuma}@cs.stanford.edu

## Abstract

Pretrained language models have achieved state-of-the-art performance when adapted to a downstream NLP task. However, theoretical analysis of these models is scarce and challenging since the pretraining and downstream tasks can be very different. We propose an analysis framework that links the pretraining and downstream tasks with an underlying latent variable generative model of text — the downstream classifier must recover a function of the posterior distribution over the latent variables. We analyze head tuning (learning a classifier on top of the frozen pretrained model) and prompt tuning in this setting. The generative model in our analysis is either a Hidden Markov Model (HMM) or an HMM augmented with a latent memory component, motivated by long-term dependencies in natural language. We show that 1) under certain non-degeneracy conditions on the HMM, simple classification heads can solve the downstream task, 2) prompt tuning obtains downstream guarantees with weaker non-degeneracy conditions, and 3) our recovery guarantees for the memory-augmented HMM are stronger than for the vanilla HMM because task-relevant information is easier to recover from the long-term memory. Experiments on synthetically generated data from HMMs back our theoretical findings.

## 1   Introduction

Natural language processing (NLP) has been revolutionized by large-scale pretrained language models such as BERT [4] and GPT [25], which are adapted to a variety of downstream NLP tasks. Although a large body of empirical work seeks to understand the effectiveness of pretrained models [7, 5, 12, 35, 34, 11, 27, 15], theoretical understanding is scarce. Theoretically analyzing the relationship between the pretraining and downstream tasks is challenging because pretraining and downstream settings can greatly differ.

The key starting point for our analysis is to link the pretraining and downstream settings through an underlying generative model of the data. We model the data distribution as a latent variable model and the downstream task as a function of the latent variables. Assuming that pretraining on a large corpus allows us to learn the generative model, the conditional token probabilities predicted by the pretrained model carry information about the hidden variables. In downstream adaptation, we aim to recover this information to solve the downstream task.

Though full finetuning is the de facto empirical standard, analyzing it is challenging because it requires characterizing the weights of the pretrained model. In this paper, we focus on *head tuning*

35th Conference on Neural Information Processing Systems (NeurIPS 2021).

and prompt tuning, which both freeze all pretrained parameters and allow us to treat the pretrained model as a black box. Head tuning [22] trains task-specific heads on top of the pretrained model outputs. Prompt tuning [31, 19, 9, 21] optimizes a task-specific "prompt" that is concatenated to the model input. Studying prompt tuning is particularly interesting since it can match the performance of full finetuning with less computation time [19, 9, 21].

Our work contrasts with prior theoretical work [28], which *assumes* that downstream labels are recoverable via a linear head applied to the conditional token probabilities, and analyze how errors in pretraining or model misspecification propagate downstream. We consider specific generative distributions for which we can *prove* these assumptions, showing that head and prompt tuning can recover the downstream labels.

Our analysis considers two data-generating distributions with increasing realism. First, we consider data generated from a Hidden Markov Model (HMM), where the downstream task is to learn a linear classifier on the posterior distribution over the hidden states (Section 3). We prove that, under strong non-degeneracy conditions on token emission probabilities, a linear head applied to a pretrained model $G$ which outputs exact conditional token probabilities ($G_i(x) = P[X_i \,|\, x_{-i}]$) can recover the downstream label (Theorem 3.3). Furthermore, we can prove better recovery guarantees with relaxed non-degeneracy assumptions (Assumption 3.1) by using continuous prompt tuning (Theorem 3.6), reflecting the strong empirical performance of prompt tuning [19, 9, 21]. Intuitively, prompt tuning conditions the latent variables so that nonessential information for the downstream task can be ignored during the tuning phase, making task-essential information easier to recover.

Second, we also strengthen our analysis by leveraging additional structure in the data. Motivated by long-range dependences in natural language, we analyze HMM variants with additional latent "memory" variables that can store long-term information more easily than vanilla HMMs (Section 4). Here, the downstream task is to learn a linear classifier on the posterior distribution of the memory variables. We show that, under weaker non-degeneracy conditions than the first setting, an attention-based classification head can recover ground-truth downstream labels from pretrained model outputs (Theorem 4.3). Intuitively, our recovery guarantees improve because the classification head can focus on the persistent, task-essential information in the memory while ignoring other transient and nonessential aspects of the latent variables. As with the vanilla HMM, we analyze prompt tuning for relaxing the non-degeneracy conditions even further (Theorem 4.6).

In summary, we relate the pretraining and downstream tasks by assuming that the downstream task is to learn a classifier on the posterior distributions of the latent variables defined by an underlying generative model of text. Our theoretical contributions are: 1) in this setting we analyze an HMM generative model show that simple classification heads can recover the true downstream labels under certain non-degeneracy assumptions, 2) we prove that soft prompt tuning can relax the non-degeneracy assumptions needed for downstream recovery making it easier to extract task-specific information, and 3) our recovery guarantees are stronger for memory-augmented HMMs in comparison to the vanilla HMM when tuning an attention-based classfication head.

We empirically evaluate our theoretical results with language models pretrained on synthetically generated data from HMMs. We find that prompt tuning obtains good downstream performance when our non-degeneracy conditions are relaxed, whereas head tuning performs poorly. Furthermore, we show that head tuning obtains better downstream performance when data is generated from a memory-augmented HMM, compared to a vanilla HMM, as is predicted by our theory.

## 1.1 Related works

The black box nature of BERT and related models has inspired a variety of empirical works which seek to understand them. Probing papers study whether a pretrained model computes various types of structured information (e.g., syntactic [35, 11]) by evaluating the performance of simple classifiers, or probes, on the representations [7, 12, 34, 27, 15]. Other papers ablate various aspects of pretraining, such as changing the masking scheme [14, 20, 40] or permuting the word order [32].

In comparison, theoretical analysis of pretrained language models is limited. Besides [28], which we discussed in Section 1, Zhang and Hashimoto [40] analyze using a linear classifier to approximately recover the latent variable in a Gaussian graphical model with sparse dependencies between observed variables. However, their analysis and setting are focused towards understanding syntactic dependencies between tokens, whereas we directly model and analyze downstream performance.

Prompt-based tuning [31, 19, 9, 21, 13, 6, 41, 2, 23], which has improved empirical downstream performance for lightweight adaptation methods beyond head tuning to approach full finetuning, is an important focus of our theoretical analysis. Shin et al. [31] employ task-specific prompts that are optimized over the discrete token space. Schick and Schütze [29, 30] reformulate natural language tasks as cloze-style phrases to enable few-shot learning. Subsequent methods [19, 9, 21] optimize "soft" prompts, or continuous embedding vectors. Lester et al. [19] employ soft prompts on pretrained large-scale T5 [26] models and show that as the model size increases, prompt tuning performance can eventually match finetuning. Hambardzumyan et al. [9] applies a variant of soft prompt tuning to MLM models. Li and Liang [21] propose prefix tuning, which prepends a trainable prefix embedding sequence to all layers of the transformer.

More broadly, Lee et al. [18] analyze reconstruction-based self-supervised learning methods in a general setting and show that under certain conditional independence assumptions, predicting one observed variable from another allows recovery of the latent with a linear head. Other theoretical works analyzing self-supervised or constrastive learning include [1, 10, 36, 38, 37], but they do not directly relate to our particular setting.

## 2    Formulations and notations

We analyze models pretrained on masked language modeling (MLM) objectives. Let $\mathcal{X}$ denote a finite vocabulary of input tokens, $\mathcal{X}^*$ the set of variable-length sequences of tokens, and $X = (X_1, \ldots, X_T) \in \mathcal{X}^*$ a random sequence of $T$ tokens. Let $\Delta^{|\mathcal{X}|}$ denote the space of probability distributions over tokens.

**Pretraining and downstream task.** Let $G(x) = (G_1(x), G_2(x), \ldots)$ denote the masked language model which predicts a probability vector for each timestep in the input $x$. Our theoretical abstraction is that $G_i$ perfectly computes the distribution of $X_i$, the $i$-th token, conditioned on all other tokens: $G_i(x) = P[X_i | X_{-i} = x_{-i}]$. Here $P[X_i \,|\, X_{-i} = x_{-i}] \in \Delta^{|\mathcal{X}|}$ is a probability vector. In particular, $G_i(x)$ does not depend on $x_i$. The downstream task involves labeled examples $(x, F^\star(x)) \in \mathcal{X}^* \times \mathcal{Y}$, where $F^\star : \mathcal{X}^* \to \mathcal{Y}$ provides ground-truth downstream labels and $\mathcal{Y}$ is a discrete set of labels for classification.

**Head and prompt tuning.** Head tuning trains a classification head $f$ on top of fixed model outputs, resulting in the classifier $F(x) = \mathbb{1}(f(G(x)) \geqslant 0)$. We expect $f$ to be a simple function such as a linear or one layer attention model. We also analyze variants where $f$ also takes the tokens $x$ or embeddings of $x$ as input, which provides additional information. Soft prompt tuning requires viewing the pretrained model $G$ as a function of the token embeddings; we refer to this model by $\overline{G}$. Letting $e(x) = e(x_1), \ldots, e(x_t)$ denote the token embeddings, we have $\overline{G}(e(x)) = G(x)$. Soft prompt tuning concatenates a trainable prompt $u$ so that the model output is $\overline{G}((u, e(x))$. We consider simultaneously training the prompt parameter $u$ and a classification head to fit the downstream task.

**Notations.** Let $\Delta^d$ denote the space of $d$-dimensional probability vectors. We work with discrete random variables $V$ taking values in a finite set $\mathcal{V}$. We use $P[V] \in \Delta^{|\mathcal{V}|}$ to denote the distribution of $V$ and $P[U \,|\, V = v] \in \mathbb{R}^{|\mathcal{U}|}$ the conditional distribution of $U$ given $V = v$. $\Pr(V = v) \in [0, 1]$ will denote the probability that $V$ takes values $v$. We also let $P[U = u \,|\, V] \in \mathbb{R}^{|\mathcal{V}|}$ denote the vector with entries $\Pr(U = u \,|\, V = v)$. $P[U \,|\, V] \in \mathbb{R}^{|\mathcal{U}| \times |\mathcal{V}|}$ will describe the matrix with entries $P[U \,|\, V]_{u,v} = \Pr(U = u \,|\, V = v)$.

For a sequence $v = (v_1, \ldots, v_t)$, we use the notation $v_{i:j}$ for $i \leqslant j$ to denote $(v_i, \ldots, v_j)$, and $v_{-i}$ to denote $(v_{1:i-1}, v_{i+1:t})$. We let $\mathbb{1}$ denote the indicator function. For set $\mathcal{V}$, we let $\mathcal{V}^* = \mathcal{V}^1 \cup \mathcal{V}^2 \cup \cdots$ denote variable-length sequences of elements of $\mathcal{V}$. Let $\odot$ denote elementwise product. Let $\mathbf{1}_d, \mathbf{0}_d$ denote the $d$-dimensional all-1's and all-0's vector. We omit the subscript if the dimension is clear from context. For two vectors $a, b \in \mathbb{R}^d$, we let $a/b$ denote their element-wise division. We use $\mathrm{supp}(a)$ to denote the set of indices where vector $a$ is non-zero.

## 3    Analysis for Hidden Markov Models

Defining a relation between pretraining and downstream tasks is the foremost challenge for analysis. We propose to link the two via latent variable generative assumptions on the input distribution. We

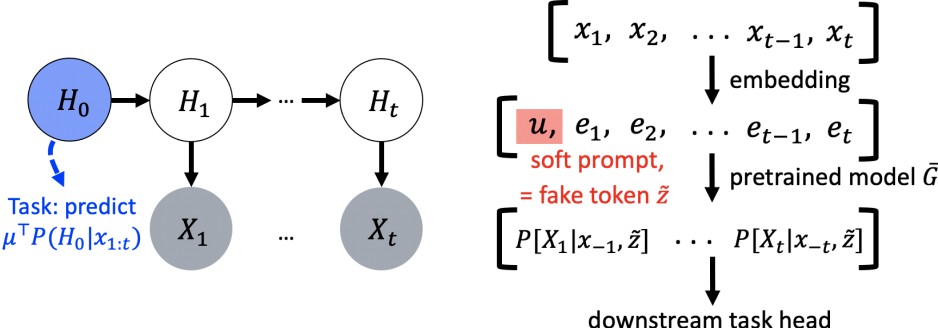

Figure 1: **Left:** Illustration of HMM graphical model. **Right:** Overview of the formulation and analysis setting for prompt (and head) tuning. To abstractify soft prompt tuning, we note that every token has a natural embedding, the corresponding row of the emission probability matrix. We view prompt tuning as adding a fake token $\tilde{z}$ to the vocabulary, assigning it a row $u$ in the emission matrix, and prepending it to the input embedding sequence. More details are provided in Section 3.1.

model the downstream task as a function of the posterior distribution of the latent variables. Towards a first result, this section studies the case where inputs are generated by HMMs (see Figure 1 (left)), which have been well-studied in the context of language and speech processing (see e.g. [24, 17, 3]).

**Data distribution.** Let $\mathcal{H}$ denote the hidden state space of the HMM. We use $H = (H_0, H_1, \ldots, H_T) \in \mathcal{H}^*$ to denote the sequence of hidden states. For all timesteps $i > 0$, the transition probabilities are time-invariant, i.e. $P[H_i \mid H_{i-1}] = A$ for $A \in \mathbb{R}^{|\mathcal{H}| \times |\mathcal{H}|}$. For each timestep $i \geqslant 1$, tokens $X_i$ are emitted following some time-invariant probability: $P[X_i \mid H_i] = W$ for $W \in \mathbb{R}^{|\mathcal{X}| \times |\mathcal{H}|}$. The joint probability of $X, H$ is

$$\Pr(X, H = x, h \mid T = t) = \Pr(H_0 = h_0) \prod_{i=1}^{t} \Pr(H_i = h_i \mid H_{i-1} = h_{i-1}) \Pr(X_i = x_i \mid H_i = h_i).$$

**Downstream tasks.** We assume that $H_0$ has the meaningful information for the downstream task, which is a binary classification task where the ground-truth labeling $F^\star$ is assumed to be a linear classifier on the posterior $P[H_0 \mid X_{1:T} = x]$:

$$F^\star(x) = \mathbb{1}(\mu^\top P[H_0 \mid X_{1:T} = x] \geqslant 0) \tag{3.1}$$

for $\mu \in \mathbb{R}^{|\mathcal{H}|}$. Our results are easily extended to the multiclass setting. We consider tuning a linear head for the downstream classifier, which formally computes $\mathbb{1}(b^\top G_1(x) \geqslant 0)$ for $b \in \mathbb{R}^{|\mathcal{X}|}$. The following non-degeneracy condition is crucial for our recovery result in this setting.

**Assumption 3.1** (Non-degeneracy, vanilla HMM). *The token emission probability matrix $W$ has linearly independent columns.*

We also require the following regularity conditions on $H_0$ and the state transitions.

**Assumption 3.2** (Regularity). *The Markov chain $H_0, H_1, \ldots$ is ergodic, and $P[H_0]$ has full support.*

We show that if $W$ has linearly independent columns, a linear head fits downstream labels.

**Theorem 3.3.** *Assume that non-degeneracy (Assumption 3.1) and regularity (Assumption 3.2) hold. Then any downstream task $F^\star(x)$ of the form (3.1) can be computed by a linear head on $G$ applied to a shifted sequence. That is, there exists linear head weights $b \in \mathbb{R}^{|\mathcal{X}|}$ such that for all $x \in \mathrm{supp}(P[X])$,*

$$F^\star(x) = \mathbb{1}(b^\top G_1(x') \geqslant 0)$$

*where $x' = (\varnothing, x_{1:t})$ is the concatenation of a special token $\varnothing$ with $x$.[1]*

The key for the proof is to leverage the following general statement about random variables $U, V, Z$ such that $U \perp V \mid Z$, which decomposes the expression for $P[U \mid V]$.

---

[1]We note that $G_1(x')$ does not depend on $x_1'$ and therefore $x_1'$ can be any token.

**Proposition 3.4.** *Let $U, V, Z$ be random variables such that $U \perp V \mid Z$. Then for any $v$, $P[U \mid V = v] = P[U \mid Z] \cdot P[Z \mid V = v]$. Thus, if $P[U \mid Z]$ has a left inverse $(P[U \mid Z])^\dagger$, then $P[Z \mid V = v] = (P[U \mid Z])^\dagger P[U \mid V = v]$.*

By the conditional independence structure of the HMM, Proposition 3.4 immediately implies

$$G_1(x') = W P[H_1 | X_{2:T+1} = x] \implies P[H_1 | X_{2:T+1} = x] = W^\dagger G_1(x')$$

where $W^\dagger$ is the left inverse for $W$, guaranteed to exist by Assumption 3.1. This lets us recover $P[H_1 | X_{2:T+1} = x]$ by applying a linear function to $G_1(x')$. Additional linear functions will be sufficient to obtain $\mu^\top P[H_0 | X_{1:T} = x]$ from $P[H_1 | X_{2:T+1} = x]$. We provide the full proof in Section B.

Proposition 3.4 is reminiscent of the arguments of [18], which leverages the independence structure in the same way. Subsequent sections will require more complicated analyses and recovery procedures.

A drawback of Theorem 3.3 is that it relies heavily on assuming $W$ has full column rank, which implies the necessary condition that $|\mathcal{H}| \leq |\mathcal{X}|$. Without this assumption, it is unclear how to recover $P[H_0 \mid X_{1:T} = x]$ from $G(x)$ alone. However, in realistic settings we would expect $|\mathcal{H}| > |\mathcal{X}|$, as increasing the size of the hidden state space improves language modeling capabilities of HMMs [3].

## 3.1 Relaxed non-degeneracy assumptions via prompt tuning

In this section, we study applying soft, or continuous, prompt tuning [19, 9] to the setting above. We show that by using soft prompt tuning, we can recover $F^\star$ using a linear head on $G$ for HMMs where the non-degeneracy assumptions on $W$ are relaxed. Our analysis provides insight into the empirical successes of prompt-tuning: intuitively, prompt tuning enables better recovery of the downstream task by conditioning the output of $G$ to only contain task-specific information.

Soft prompt tuning trains task-specific embedding vectors, but analyzing how the model processes embedding vectors is challenging because it requires opening up the black box of the pretrained model. Thus, we require additional abstractions about how the pretrained model processes the embedding vectors. We will extend the mask language model $G$ to a model $\overline{G}$ that maps a sequence of embeddings $e_1, \ldots, e_t$ to conditional probabilities $G_1(x), \ldots, G_t(x)$ as follows. We observe that each token $z$ in the vocabulary $\mathcal{X}$ naturally corresponds to a $|\mathcal{H}|$-dimensional vector: the $z$-th row of the emission probability matrix $W$, or equivalently, $P[X_i = z \mid H_i]$. We denote this embedding by $e(z)$ and call the family of embeddings $\{e(z) : z \in \mathcal{X}\}$ proper embeddings. A fundamental property of HMMs is that the conditional probability $P[X_i \mid X_{-i} = x_{-i}]$ only depends on $x_1, \ldots, x_t$ through their embeddings $e(x) = (e(x_1), \ldots, e(x_t))$. In other words, there exists a function $\overline{G}_i$ such that

$$G_i(x_1, \ldots, x_t) = \overline{G}_i(e(x_1), \ldots, e(x_t))$$

In particular, we let $\overline{G}_i$ compute the standard message passing algorithm [16] that computes the conditional probability of HMMs. This ensures that $\overline{G}_i$ is well defined on all sequences of nonnegative vectors in $[0, 1]^{|\mathcal{H}|}$, beyond sequences of proper embeddings. We assume that pretraining produces this $\overline{G}_i$, which we treat as a blackbox for prompt tuning.

In particular, for prompt tuning we can consider the case where we pass an arbitrary nonnegative vector $u \in [0, 1]^{|\mathcal{H}|}$ to $\overline{G}$ in the first argument and proper embeddings at positions $i > 1$. We can interpret $u$ as the embedding of a fake token $\widetilde{z}$. Concretely, consider adding a new token $\widetilde{z}$ to the vocabulary $\mathcal{X}$, and changing the emission probability at position 1 to satisfy $P[X_1 = \widetilde{z} \mid H_1] = u$ and for all $z \neq \widetilde{z}$, $P[X_1 = z \mid H_1] \propto (1 - u) \odot e(z)$. Then $\overline{G}_i(u, e(x_1), \ldots, e(x_t))$ precisely computes the conditional probability $P[X_i \mid X_{-i} = (\widetilde{z}, x_1, \ldots, x_t)_{-i}]$ under the modified HMM. We refer the readers to Section C for the formal definition of $\overline{G}_i$ and formal proofs of the interpretation above.

We consider a downstream training algorithm which trains the prompt tuning parameter $u$ described above and a linear classification head. Letting $u$ denote the trainable prompt parameter and $b \in \mathbb{R}^{|\mathcal{X}|}$ the trainable linear head weights, the model uses the embedding sequence

$$\widehat{e}(x) \triangleq (u, e(\varnothing), e(x_1), \ldots, e(x_t)) \tag{3.2}$$

and outputs the prediction $F(x) = \mathbb{1}(b^\top G_2(\widehat{e}(x)) \geq 0)$. We can provide recovery guarantees for this model if the ground-truth classifier weights $\mu$ (defined in (3.1)) and columns of the HMM transition matrix $A$ satisfy the following relaxation of the requirement in Theorem 3.3 that $W$ is nondegenerate.

**Assumption 3.5** (Relaxed non-degeneracy condition). *There exists a set of essential hidden states $\mathcal{H}^\star \subseteq \mathcal{H}$, so that the columns of $W$ corresponding to $\mathcal{H}^\star$, $\{W_{:,h}\}_{h \in \mathcal{H}^\star}$, are linearly independent. Furthermore, $\mathcal{H}^\star$ covers all meaningful information for the downstream tasks:* $\mathrm{supp}(\mu) \subseteq \mathcal{H}^\star$.

*In addition, a last technical requirement on $\mathcal{H}^\star$ is as follows: there exists a set $\mathcal{B} \subseteq \mathcal{H}$ such that $\mathcal{H}^\star = \cup_{h \in \mathcal{B}} \mathrm{supp}(A_{:,h})$. In other words, $\mathcal{H}^\star$ must be the set of all states reachable by starting from some state in $\mathcal{B}$ and transitioning one step in the hidden Markov chain.*

Compared to Assumption 3.1, which required that *all* columns of $W$ are linearly independent, Assumption 3.5 only requires linear independence on a subset $\mathcal{H}^\star$ of essential states. In the setting where $|\mathcal{H}| > |\mathcal{X}|$, the condition for Theorem 3.3 can never hold. On the other hand, Assumption 3.5 could still hold, for example, if $|\mathrm{supp}(\mu)| < |\mathcal{X}|$ and the set of columns of $W$ corresponding to hidden states in $\mathrm{supp}(\mu)$ is linearly independent. The last technical requirement in Assumption 3.5 is also required, which could be satisfied if columns of $A$ are sparse. The following theorem shows that when Assumption 3.5 holds, we can recover $F^\star$ using soft prompt tuning with a linear head.

**Theorem 3.6.** *In the above setting, assume that Assumptions 3.2 and 3.5 hold. Then $F^\star$ can be computed using soft prompt tuning with a linear head on $\overline{G}$. Concretely, there is a continuous prompt parameter $u \in \mathbb{R}^{|\mathcal{H}|}$ and weight vector $b \in \mathbb{R}^{|\mathcal{X}|}$, such that for all $x \in \mathrm{supp}(P[X])$,*

$$F^\star(x) = \mathbb{1}(b^\top \overline{G}_2(\widehat{e}(x)) \geq 0)$$

*where $\widehat{e}$ prepends $u$ to the input embedding sequence, as defined in* (3.2).

Theorem 3.6 provides a stronger recovery result than Theorem 3.3, which only used a linear head. This is also reflected in our synthetic experiments (Section 5), and prior work which shows that variants of prompt tuning can perform much better than only training the last few layers of the model [21]. Our theory suggests that prompt tuning could help by conditioning the hidden variables to remove nonessential information for the task from the output of $G$. This makes task-essential information easier to recover.

The key proof intuition is that although recovering $P[H_0 \mid X_{1:T} = x]$ is impossible without strong non-degeneracy conditions (Assumption 3.1), we can aim to recover $P[H_0 \mid X_{1:T} = x]$ on the subset of essential states $\mathcal{H}^\star$ defined in Assumption 3.5, which suffices for computing $\mu^\top P[H_0 \mid X_{1:T} = x]$, since $\mathcal{H}^\star \supseteq \mathrm{supp}(\mu)$. To recover $P[H_0 \mid X_{1:T} = x]$ on $\mathcal{H}^\star$, we observe in Lemma C.2 that prepending the prompt $u$ is equivalent to introducing a modified random sequence $\widehat{X}$ and fake token $\widetilde{z}$ which influences the posterior of $H_2$ as follows:

$$\overline{G}_2(\widehat{e}(x)) = r_x W D(P[H_2 \mid \widehat{X}_1 = \widetilde{z}] \odot P[H_0 \mid X_{1:T} = x]) \tag{3.3}$$

for invertible diagonal matrix $D$ and positive scalar $r_x$. We choose $u$ so $P[H_2 \mid \widehat{X}_1 = \widetilde{z}] \odot P[H_0 \mid X_{1:T} = x]$ is supported only on $\mathcal{H}^\star$. As corresponding columns of $W$ are linearly independent (Assumption 3.5), we recover $\mathrm{Pr}(H_0 = h \mid X_{1:T} = x)$ for $h \in \mathcal{H}^\star$ via a linear function of $\overline{G}_2(\widehat{e}(x))$. This suffices for computing $\mu^\top P[H_0 \mid X_{1:T} = x]$. For more details, see Section C.

## 4 Analysis for memory-augmented Hidden Markov Models

We study a memory-augmented HMM which explicitly disentangles the evolution of hidden states from a persistent "memory" variable. Inspired by natural sentences, this model is intended to better capture the distinction between syntax, which constantly evolves, and semantics, which changes less. This additional structure in the generative model allows us to strengthen our results by relaxing the non-degeneracy conditions on $W$, the token emission probabilities. Thus, both head and prompt tuning are more powerful in this setting compared to Section 3 and can recover the downstream label with weaker non-degeneracy assumptions on $W$. In Section 4.2, we show that soft prompt tuning also provides an advantage over head tuning alone.

**Data distribution.** The memory-augmented HMM, depicted in Figure 2, can be viewed as a generative variant of memory networks [39, 33] and is closely related to Hidden Topic Markov Models [8]. There are two sets of latent variables in the memory-augmented HMM: a Markov chain on hidden states $H_0, H_1, \ldots$, meant to model the evolution of syntax, and a persistent "memory" $M = (M_1, \ldots, M_N)$ with $N$ total cells, where each $M_i$ takes values in a finite set $\mathcal{M}$. The full joint

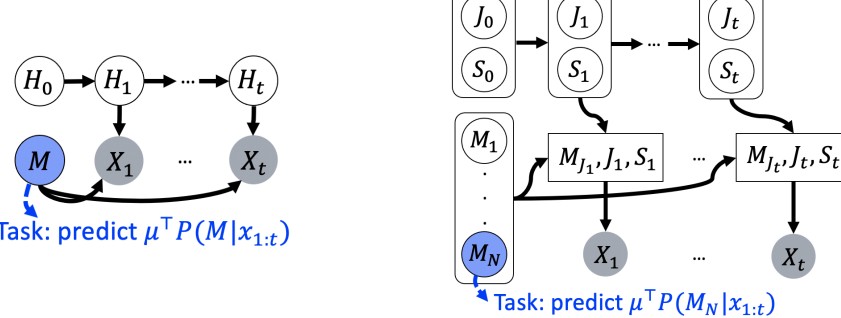

Figure 2: **Left:** Memory-augmented HMM with a single memory cell. The memory $M$ and hidden state $H_i$ determine the emission probabilities for each state $X_i$. **Right:** Memory-augmented HMM with multiple memories $M_1, \ldots, M_N$. The hidden state $H_i$ consists of a cell index $J_i$ and syntax state $S_i$. To sample $X_i$, we first look up the $J_i$-th memory cell $M_{J_i}$. The token emission probability is then determined by the tuple $(M_{J_i}, J_i, S_i)$.

probability is as follows:

$$\Pr(X, H, M = x, h, m | T = t) =$$

$$\Pr(M = m)\Pr(H_0 = h_0) \prod_{i=1}^{t} \Pr(H_i = h_i | H_{i-1} = h_{i-1})\Pr(X_i = x_i | M = m, H_i = h_i)$$

The hidden state is modified to explicitly consist of a disentangled cell index $J \in [N]$ and syntax state $S \in \mathcal{S}$, such that $H_i = (J_i, S_i)$ and $\mathcal{H} = [N] \times \mathcal{S}$. To sample the token at timestep $i$ given the hidden state $H_i = (J_i, S_i)$, we first use $J_i$ to index the memory $M$, obtaining the random variable $M_{J_i}$. $X_i$ is then sampled according to some time-invariant probability depending on $M_{J_i}, J_i, S_i$:

$$P[X_i \,|\, M = m, H_i = (j, s)] = P[X_i \,|\, M_{J_i} = m_j, H_i = (j, s)] = W_{:,(m_j, j, s)}$$

Here $W \in \mathbb{R}^{|\mathcal{X}| \times |\mathcal{M}||\mathcal{H}|}$ stores the emission probabilities for each choice of memory cell value and hidden state. Note that in particular, the conditional probabilities for $X_i$ only depend on a single memory cell for each timestep. We also note that memory-augmented HMMs can be viewed as vanilla HMMs with structured transitions because $(H_0, M), (H_1, M), \ldots$ can be viewed as a Markov chain where the memory component does not change.

**Example 4.1** (Generating natural sentence with memory-augmented HMM). *We consider how this model may generate the sentence "The cow in the pasture rolled on the grass' happily." $M_1$ could store the subject ("cow"), $M_2$ the location ("pasture"), $M_3$ the sentiment ("happily"), and $S_i$ could determine part-of-speech. For timesteps where "cow" and "rolled" are emitted $J_i = 1$ because we emit information related to the sentence subject. Timesteps for "pasture" and "grass" have $J_i = 2$.*

**Downstream tasks.** We consider downstream tasks where ground-truth labels are obtained via a linear classifier on the posterior distribution of a particular memory cell $j^\star \in [N]$: $F^\star(x) = \mathbb{1}(\mu^\top P[M_{j^\star} | X_{1:T} = x] \geqslant 0)$, where $\mu \in \mathbb{R}^{|\mathcal{M}|}$. Intuitively, this formulation models downstream tasks which depend on a particular aspect of the semantics but not on syntax (e.g. in the setting of Example 4.1, if $j^\star = 3$, the task is sentiment analysis).

### 4.1 Tuning attention head for recovering ground-truth downstream labels

To recover the downstream labeling, we require an attention-based classification head, which is a function of both the input embeddings and outputs of $G$. Formally, let $q \in \mathbb{R}^{|\mathcal{H}|+1}$ denote a query parameter and $\beta_1, \ldots, \beta_t \in \mathbb{R}^{|\mathcal{H}|+1}$ denote trainable position embeddings. Given pretrained model outputs $G_i(x)$ and trainable token embeddings $e(x_i)$, the attention head Attn($\cdot$) applies key and value functions $K, V$ to compute the output as follows:

$$\mathcal{I} \triangleq \arg\max_{i}\{q^\top\big(K(G_i(x)) + \beta_i\big)\} \tag{4.1}$$

$$\text{Attn}((G_i(x), e(x_i))_{i=1}^{t}) \triangleq \frac{1}{|\mathcal{I}|} \sum_{i \in \mathcal{I}} V(G_i(x), e(x_i)) \tag{4.2}$$

where $\arg\max$ refers to the set of indices achieving the maximum in (4.1). We note that standard attention heads in practice rely on the softmax function, but the expression based on $\arg\max$ above captures the limiting behavior as $\|q\|_2 \to \infty$. We consider linear key functions given by $K(G_i(x)) = \Theta^{(K)}G_i(x)$. The value function $V : \mathbb{R}^{|\mathcal{X}|} \times \mathbb{R}^{|\mathcal{M}||\mathcal{H}|} \to \mathbb{R}$ uses parameters $\Theta^{(V)} \in \mathbb{R}^{|\mathcal{M}||\mathcal{H}| \times |\mathcal{X}|}$ and $b \in \mathbb{R}^{|\mathcal{M}||\mathcal{H}|}$ and computes $V(G_i(x), e(x_i)) = b^\top((\Theta^{(V)}G_i(x)) \odot e(x_i))$.

Because our generative model disentangles $H$ and $M$, we can relax the non-degeneracy assumption on the token emission probabilities $W$, compared to Theorem 3.3. The relaxed assumption only requires the columns $\{W_{:,(m,h)}\}_{m\in\mathcal{M},h\in\mathcal{H}^\star}$ to be linearly independent in a subset $\mathcal{H}^\star$ of "recoverable" hidden states, whereas Assumption 3.1 required all columns to be linearly independent.

**Assumption 4.2** (Existence of "recoverable" hidden states). *There exists a set of recoverable hidden states $\mathcal{H}^\star = \{j^\star\} \times \mathcal{S}^\star$, such that the collection of token emission probabilities from $\mathcal{M} \times \mathcal{H}^\star$, $\{W_{:,(m,h)}\}_{m\in\mathcal{M},h\in\mathcal{H}^\star}$, is a linearly independent set of vectors.*

*Furthermore, the span of these vectors must be disjoint from the span of token emission probabilities from $\mathcal{M} \times (\mathcal{H}\backslash\mathcal{H}^\star)$: $\mathrm{span}(\{W_{:,(m,h)}\}_{m\in\mathcal{M},h\in\mathcal{H}^\star}) \cap \mathrm{span}(\{W_{:,(m,h')}\}_{m\in\mathcal{M},h\in\mathcal{H}\backslash\mathcal{H}^\star}) = \{\mathbf{0}_{|\mathcal{X}|}\}$.*

Note that the non-degeneracy condition of Theorem 3.3 would require $\{W_{:,(m,h)}\}_{m\in\mathcal{M},h\in\mathcal{H}}$ to be linearly independent, whereas Assumption 4.2 only requires linear independence for $h \in \mathcal{H}^\star$. The second condition states that $\mathcal{H}^\star$ and $\mathcal{H}\backslash\mathcal{H}^\star$ are distinguishable by the token emission probabilities.

We explain Assumption 4.2 in the setting of Example 4.1. For natural language, there might be choices of $h = (j_i, s_i)$ for which the set $\{W_{:,(m,h)}\}_{m\in\mathcal{M}}$ of token emission probabilities is fundamentally not very diverse, and therefore not linearly independent. For example, if the syntax $s_i$ indicates "article", i.e. words such as "a", "an", and "the", the token emission probabilities would carry little information about $M_{j_i}$ because the choice of article does not depend much on semantics, so columns corresponding to $s_i = $ "article" would not be linearly independent, violating Assumption 3.1. However, Assumption 4.2 allows us to avoid this issue by placing such $h$ in $\mathcal{H}\backslash\mathcal{H}^\star$, a set of hidden states which we can ignore, and only including hidden states which carry a lot of information about $M$ in $\mathcal{H}^\star$. In Example 4.1, when $J_i = 2$ (location), $S_i = $ "noun", the position $i$ should convey a lot about the location (in this case, "pasture"), so it is more reasonable to assume that $\{W_{:,m,h}\}_{m\in\mathcal{M}}$ is linearly independent for this hidden state.

Thus, our aim is to focus on recovering information for the downstream task from positions $i$ where $H_i \in \mathcal{H}^\star$. Formally, we define the following set of input sequences containing positions $i$ where the posterior of $H_i$ given $x_{-i}$ concentrates on $\mathcal{H}^\star$:

$$\mathcal{R} \triangleq \{(x_1, \ldots, x_t) \in \mathrm{supp}(P[X]) : \exists i \text{ with } \mathrm{supp}(P[H_i \,|\, X_{-i} = x_{-i}]) \subseteq \mathcal{H}^\star\} \qquad (4.3)$$

The following theorem shows that under Assumption 4.2, we can recover $F^\star$ using the attention head described above, if $x \in \mathcal{R}$ is nonempty. Note that $\mathcal{R}$ is nonempty if the posterior of $H_i$ concentrates on $\mathcal{H}^\star$ for some $i$. For natural language, it is realistic to assume this can occur because syntactic aspects of a sentence are typically low-entropy when the full sentence is observed.

**Theorem 4.3.** *Assume that non-degeneracy (Assumption 4.2) and regularity (Assumption 3.2) hold. Define $\mathcal{R}$ as in (4.3). Then there exist an attention head on $G(x)$ and token embeddings $e(x_i)$ such that the following holds for any $x \in \mathcal{R}$:*

$$F^\star(x) = \mathbb{1}(\mathrm{Attn}((G_i(x), e(x_i))_{i=1}^t) \geqslant 0)$$

*where the function $\mathrm{Attn}$ is in the form described in (4.2).*

The idea is to use the attention mechanism to attend to positions $i$ where $\mathrm{supp}(P[H_i \,|\, X_{-i} = x_{-i}]) \subseteq \mathcal{H}^\star$. The intuition of Assumption 4.2 is that such positions are more informative for recovering the latent posteriors; indeed, from the outputs $G_i(x)$ at such $i$, the value function in the attention will be able to recover $P[M_{j^\star} \,|\, X_{1:T} = x]$. A full proof is provided in Section D.1.

## 4.2 Guarantees for prompt-tuning

Though the generative modeling assumptions in this section already allowed relaxed non-degeneracy assumptions, applying soft prompt tuning allows us to relax them even further. For simplicity, we consider the setting where there is a single memory cell, so $M \in \mathcal{M}$, and the downstream task is a linear classifier on the posterior of the memory: $F^\star(x) = \mathbb{1}(\mu^\top P[M|X_{1:T} = x] \geqslant 0)$. This

simplified setting doesn't require the explicit disentanglement between $J_i$ and $S_i$ in $H_i$. We analyze continuous prompt-tuning in a setting where the pretrained model $\overline{G}$ follows the same abstraction as in Section 3.1. We modify the model to take $|\mathcal{M}||\mathcal{H}|$-dimensional vectors, so the proper embedding for token $z$ is given by $e(z) = P[X_i = z|M, H_i] = W_{z,:}^\top$. In Section D.3, we describe the formal construction and interpretation of $\overline{G}$ in the more general setting with more memories.

Letting $u \in \mathbb{R}^{|\mathcal{M}||\mathcal{H}|}$ denote the trainable prompt parameter, we define the input embeddings

$$\widehat{e}(x) \triangleq (u, e(x_1), \ldots, e(x_t)) \tag{4.4}$$

The downstream model applies an attention head to the output of $\overline{G}$: $F(x) = \mathbb{1}(\text{Attn}((\overline{G}_i(\widehat{e}(x)), \widehat{e}_i(x))_{i=1}^{t+1}) \geqslant 0)$, where Attn is defined in (4.2). An additional stationarity assumption on $P[H_0]$ will simplify the recovery procedure (though it can be removed).

**Assumption 4.4** (Stationarity). *Assumption 3.2 holds on the Markov chain $H_0, H_1, \ldots$. Furthermore, $P[H_0]$ is the stationary distribution: $P[H_0] = AP[H_0]$, where $A$ is the transition matrix.*

As before, we assume sparsity of $\mu$ and some non-degeneracy of $W$, though the assumption is more relaxed and easier to state compared to the vanilla HMM setting.

**Assumption 4.5** (Relaxed version of Assumption 4.2). *Let $\mathcal{M}^\star \triangleq \text{supp}(\mu)$ denote the set of non-zero coordinates in $\mu$. There exists a set of recoverable hidden states $\mathcal{H}^\star$, such that the collection of token emission probabilities from $\mathcal{M}^\star \times \mathcal{H}^\star$, $\{W_{:,(m,h)}\}_{m \in \mathcal{M}^\star, h \in \mathcal{H}^\star}$, is linearly independent.*

*Furthermore, the span of these vectors must be disjoint from the span of token emission probabilities from $\mathcal{M}^\star \times (\mathcal{H} \backslash \mathcal{H}^\star)$: $\text{span}(\{W_{:,(m,h)}\}_{m \in \mathcal{M}^\star, h \in \mathcal{H}^\star}) \cap \text{span}(\{W_{:,(m,h')}\}_{m \in \mathcal{M}^\star, h \in \mathcal{H} \backslash \mathcal{H}^\star}) = \{\mathbf{0}_{|\mathcal{X}|}\}$.*

We note that Assumption 4.5, and Assumption D.5 for multiple memories, are relaxations of Assumption 4.2, as they only consider memory values in $\text{supp}(\mu)$, whereas Assumption 4.2 considers all $m \in \mathcal{M}$. An additional advantage of the memory-augmented HMM is that Assumption 4.2 is simpler than Assumption 3.1 and does not require any conditions on the transition matrix $A$. We now state our result for recovering $F^\star$ with soft prompt tuning and an attention head.

**Theorem 4.6.** *In the setting above, suppose that non-degeneracy Assumption 4.5 and stationarity Assumption 4.4 hold. Then there exists a prompt $u$ and attention head on $\overline{G}(\widehat{e}(x))$ and the token embeddings which can compute the ground-truth $F^\star(x)$ for any $x \in \mathcal{R}$, defined in (4.3):*

$$F^\star(x) = \mathbb{1}(\text{Attn}((\overline{G}_i(\widehat{e}(x)), \widehat{e}_i(x))_{i=1}^{t+1}) \geqslant 0)$$

*where $\widehat{e}$ is the embedding in (4.4) and Attn is defined in (4.2).*

The intuition for this proof is similar to Theorem 3.6: the soft prompt conditions the memory $M$ to concentrate on $\text{supp}(\mu)$. As a result, all irrelevant information to the task is removed from $\overline{G}_i(\widehat{e}(x))$, making it easier to recover the task-specific information about the posterior of $M$. A more general theorem statement for the multiple memories setting, and the full proof, is provided in Section D.3

## 5  Simulations

We empirically evaluate our theoretical results by pretraining a BERT-like masked language model (MLM) [4] on synthetic data generated by an HMM. Our goal is to verify key implications of our theory in a more realistic setting where some assumptions, such as that $G$ outputs exact conditional probabilities, may not hold. First, we compare head and prompt tuning and show that prompt tuning improves downstream performance, especially when the recovery problem is degenerate. Second, we compare the effect of changing the data distribution from vanilla HMMs to memory-augmented HMMs on head tuning with an attention layer. We find that the downstream performance improves when the data has a long-term memory component. These observations support our theory.

**Pretraining data and downstream task.** We generate pretraining data from an HMM with randomly generated transition matrix, emission probabilities, and start distributions. In all experiments, the HMMs have 10 vocabulary symbols, while the hidden state size varies. The downstream task uses input sequences $X_{1:T}$ of length 129, where the first token $X_1 = $ [MASK]. We consider binary classifcation where labels are generated using linear functions of the analytically-computed posteriors in the HMMs. In all experiments, the ground truth linear weight is sparse with 6 nonzero entries at uniformly random locations with Gaussian values. More details are in Appendix E.

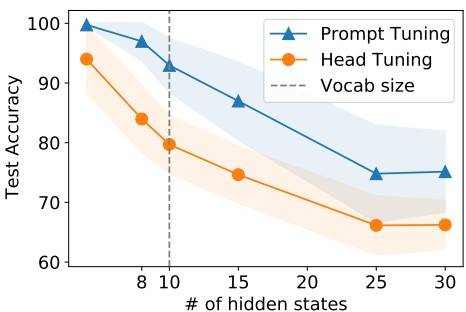 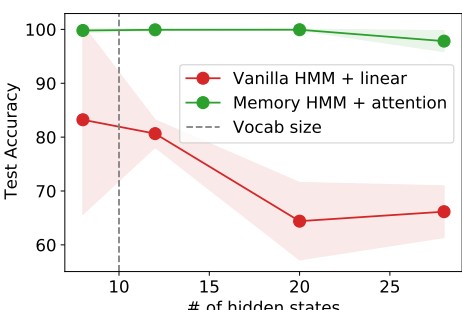

Figure 3: **Left:** Head vs. prompt tuning with a linear head on synthetically-generated HMM data, with varying hidden state sizes. Prompt tuning improves downstream accuracy especially when the problem is degenerate ($|\mathcal{H}| > |\mathcal{X}|$). **Right:** Downstream accuracy of head tuning on data from vanilla HMM vs. memory-augmented HMM, across varying values of $|\mathcal{M}||\mathcal{H}|$. Long-term dependencies in the memory-augmented HMM data improve downstream recovery with attention. We average over 20 trials (left) and 5 trials (right) of pretraining and finetuning, with 95% intervals shown.

**Head vs. prompt tuning.** We compare head and prompt tuning as the hidden state size of the HMM varies. The downstream label is computed via $\mu^\top P[H_1 \mid X_{-1} = x_{-1}]$, where $\mu$ is a random ground-truth linear weight. Head tuning learns a linear head on top of the softmax probabilities predicted by the pretrained model for filling in the first [MASK] token. Prompt tuning uses the same setup but also optimizes a length 20 continuous embedding prepended to the input sequence.

Figure 3 (left) shows that prompt tuning improves downstream performance substantially across all hidden state sizes ({4,8,10,15,25,30}). Prompt tuning improves especially when the hidden state size increases beyond the vocabulary size, which makes the recovery problem degenerate. Thus, as suggested by Theorem 3.6, prompt tuning helps relax the non-degeneracy conditions.

**Memory-augmented HMMs.** We investigate the effect of augmenting the data-generating HMM with a long-term memory. We consider the single memory case with $|\mathcal{H}| = 4$ and varying memory sizes $|\mathcal{M}| \in \{2, 3, 5, 7\}$. The downstream label is generated by computing $\mu^\top P[M \mid X_{-1} = x_{-1}]$, where $\mu$ denotes the ground-truth weights. Viewing the memory HMM as a HMM where the component on $\mathcal{M}$ never changes, we can compare against the vanilla HMMs from the previous setting. For the memory-augmented HMM, we use head tuning with a single-cell attention layer on the entire sequence of softmax probability outputs. For the vanilla HMM in the comparison, we use a linear head on the output at the first position, as an attention head would perform worse since the downstream task depends only on $H_1$ and not any other timesteps.

Figure 3 (right) verifies that head tuning recovers the downstream task better when there is more structure in the data, as predicted by Theorem 4.3. Head tuning achieves near 100% downstream accuracy on all hidden state sizes.

## 6 Conclusion

We analyze how pretraining on generic language modeling tasks can improve performance on diverse downstream tasks. In our analysis framework, the downstream task requires predicting properties of the posterior distribution over latent variables in an underlying generative model. When the generative model is a standard HMM, downstream recovery is possible with a simple classification head under strong non-degeneracy assumptions. We also show that we can relax the non-degeneracy conditions by changing the generative model to a memory-augmented HMM or using prompt tuning. The distributions studied here are meant to provide a first-cut result – we also expect similar theorems to hold for other generative models, which we leave as an interesting direction for future work.

## Acknowledgements

We thank Percy Liang, Tianyi Zhang, and Nelson Liu for helpful discussions.

## Funding statement

CW was supported by a NSF Graduate Research Fellowship. SMX was supported by a NDSEG Fellowship. TM acknowledges support of Google Faculty Award, NSF IIS 2045685, and JD.com. Additional revenue: CW received an honorarium for a talk at G-Research.

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
