## A   Limitations and societal impact

**Limitations.** The primary limitation of our work is that the analysis for prompt tuning has to be performed in an idealized setting where we assume the model processes the token embeddings in a very specific way, i.e., it treats the token embeddings as a vector describing token emission probabilities and performs message passing. Without making this assumption, it's extremely challenging to analyze the embeddings of the model directly, as this would require opening up the black box of the model. It seems unlikely that models pretrained in practice will entirely fit the assumptions we require for our prompt tuning analysis. However, we hope that our analysis can still provide insights into why prompt tuning might help, even with the assumptions it requires.

Another limitation is that our results assume the model outputs the perfect conditional token probabilities, but in practice we likely run into issues like model misspecification and suboptimal training. We believe it's possible for future work to use existing techniques to provide guarantees which account

for errors in the probabilities predicted by the model. We decided to omit such analyses in order to obtain a cleaner first result.

**Negative societal impact.** We don't foresee any negative impact of work, as our paper is theoretical and does not propose new algorithms.

# B  Proofs for Section 3

We provide the formal proof of Theorem 3.3 based on the sketch in Section 3. The following lemma will be useful in our analysis.

**Claim B.1.** *In the setting of Section 3, suppose that Assumption 3.2 holds. Fix any timestep $i \geqslant 1$. Then there exists a diagonal matrix $D$ such that for all $x \in \mathrm{supp}(P[X])$,*

$$P[H_i \,|\, X_{i+1:T+i} = x] = r_x D P[H_0 \,|\, X_{1:T} = x]$$

*where $r_x > 0$ is a positive scalar.*

*Proof.* First, we note that by Assumption 3.2, $P[H_i]$ has full support. As a consequence, $\Pr(X_{i+1:t+i} = x) > 0$. By Bayes' rule,

$$
\begin{aligned}
P[H_i \,|\, X_{i+1:T+i} = x] &= \frac{P[X_{i+1:T+i} = x \,|\, H_i] \odot P[H_i]}{\Pr(X_{i+1:T+i} = x)} \\
&= \frac{P[X_{1:T} = x \,|\, H_0] \odot P[H_0]}{\Pr(X_{i+1:T+1} = x)} \odot \frac{P[H_i]}{P[H_0]} \\
&\qquad\qquad\qquad\qquad \text{(by Markovian property of HMMs)} \\
&= P[H_0 \,|\, X_{1:T} = x] \odot \frac{P[H_i]}{P[H_0]} \cdot \frac{\Pr(X_{1:T} = x)}{\Pr(X_{i+1:T+i} = x)}
\end{aligned}
$$

Note that the vector $\frac{P[H_i]}{P[H_0]}$ has finite and positive entries. The same applies to the ratio $r_x \triangleq \frac{\Pr(X_{1:T} = x)}{\Pr(X_{i+1:T+i} = x)}$. Thus, we get the desired statement. $\qquad\square$

The proof of Theorem 3.3 follows below.

*Proof of Theorem 3.3.* By definition, $G_1(x') = P[X_1 \,|\, X_{2:T+1} = x]$. Therefore, our goal is to rewrite $P[H_0 \,|\, X_{1:T} = x]$ as a linear function of $P[X_1 | X_{2:T+1} = x]$ (up to a scaling which won't affect the linear head prediction). Concretely, we will show

$$P[H_0 \,|\, X_{1:T} = x] = r_x B P[X_1 \,|\, X_{2:T+1} = x] \tag{B.1}$$

for a scalar $r_x \geqslant 0$. With this equation, taking $b = \mu^\top B$ will give the desired result.

First, observe that $P[X_1 \,|\, X_{2:T+1} = x] = W P[H_1 \,|\, X_{2:T+1} = x]$ by Proposition 3.4. Next, we apply Claim B.1 to obtain an invertible matrix $D$ such that for all $x \in \mathrm{supp}(P[X])$, $P[H_1 | X_{2:T+1} = x] = r_x D P[H_0 | X_{1:T} = x]$, where $r_x > 0$ is a scalar.

If $W$ has full row rank, it has a left inverse $W^\dagger$ with $W^\dagger W = I_{|\mathcal{H}| \times |\mathcal{H}|}$. Choosing $b = \mu D^{-1} W^\dagger$, we obtain

$$
\begin{aligned}
\mathbb{1}(b^\top G_1(x') \geqslant 0) &= \mathbb{1}(\mu^\top D^{-1} W^\dagger W P[H_1 \,|\, X_{2:T+1} = x] \geqslant 0) \\
&= \mathbb{1}(\mu^\top P[H_0 \,|\, X_{1:T} = x] \geqslant 0) = F^\star(x)
\end{aligned}
$$

$\qquad\square$

Next, we complete the proof of Proposition 3.4.

*Proof of Proposition 3.4.* We write

$$
\begin{aligned}
P[U \mid V = v] &= \sum_z P[U, Z = z \mid V = v] \\
&= \sum_z P[U \mid Z = z, V = v]\Pr(Z = z \mid V = v) && \text{(by Bayes' rule)} \\
&= \sum_z P[U \mid Z = z]\Pr(Z = z \mid V = v) && \text{(since } U \perp V \mid Z) \\
&= P[U \mid Z]P[Z \mid V = v]
\end{aligned}
$$

$\square$

## C   Formal abstraction for prompt tuning and proofs for Section 3.1

We first formalize the definition of the model $\overline{G}$ described in Section 3.1. The model $\overline{G}$ takes a sequence of embedding vectors $v = (v_1, \dots, v_t)$ as input and implements message passing to compute a sequence of $t$ outputs. We first define left and right messages $\overleftarrow{\delta}_{i+1 \to i}(v)$ and $\overrightarrow{\delta}_{i-1 \to i}(v)$ for $i \in [t]$, as follows:

$$
\begin{aligned}
\overleftarrow{\delta}_{t+1 \to t}(e) &= P[H_t] \\
\overleftarrow{\delta}_{i \to i-1}(e) &= P[H_{i-1} \mid H_i](\overleftarrow{\delta}_{i+1 \to i}(v) \odot v_i) \; \forall 1 < i < t \\
\overrightarrow{\delta}_{0 \to 1}(e) &= P[H_1] \\
\overrightarrow{\delta}_{i \to i+1}(e) &= P[H_{i+1} \mid H_i](\overrightarrow{\delta}_{i-1 \to i}(v) \odot v_i) \; \forall 1 < i < t
\end{aligned}
$$

Next, we define the aggregated message at timestep $i$ by

$$
\tau_i(v) \triangleq
\begin{cases}
\overleftarrow{\delta}_{2 \to 1}(v) & \text{if } i = 1 \\
\dfrac{\overleftarrow{\delta}_{i+1 \to i}(v) \odot \overrightarrow{\delta}_{i-1 \to i}(v)}{P[H_i]} & \text{if } 1 < i < t \\
\overrightarrow{\delta}_{t-1 \to t}(v) & \text{if } i = t
\end{cases}
\tag{C.1}
$$

Note that if Assumption 3.2 holds about the Markov chain $H_0, H_1, \dots,$ $\tau_i(v)$ is always well-defined because $P[H_i]$ will have full support. Note that for the proper embeddings $e(x_i) = P[X_i = x_i \mid H_i]$, where for $x = (x_1, \dots, x_t)$, we use $e(x) = (e(x_1), \dots, e(x_t))$, we can check via classical results on message passing [16] that

$$
\tau_i(e(x)) = P[H_i, X_{-i} = x_{-i}]
$$

Finally, we let the model model $\overline{G}$ compute

$$
\overline{G}_i(v) = W \frac{\tau_i(v)}{\|\tau_i(v)\|_1}
$$

There is an edge case where the demoninator is 0, i.e. $\|\tau_i(v)\|_1 = 0$. To make the behavior of $\overline{G}$ well-defined, in this case we set $\overline{G}_i(v) = \mathbf{0}_{|\mathcal{X}|}$. We observe that if the input embedding are obtained by $e(x)$, $\overline{G}_i(v)$ indeed computes the desired conditional probability vector for $x \in \operatorname{supp}(P[X])$:

$$
\overline{G}_i(e(x)) = P[X_i \mid X_{-i} = x_{-i}]
$$

### C.1   Proof of Theorem 3.6

First we formalize the observation that soft prompt tuning is equivalent to adding a fake token $\widetilde{z}$ to the vocabulary with emission probabilities at timestep 1 given by $u$, and letting $\overline{G}$ compute conditional probabilities for this new distribution over sequences.

**Lemma C.1.** *In the setting of Theorem 3.6, fix any prompt vector $u \in [0, 1]^{|\mathcal{H}|}$. Define the random variable $\widehat{X}$ with the same emission probabilities as $X$ for $i > 1$: $P[\widehat{X}_i \mid H_i] = P[X_i \mid H_i]$. For timestep 1, we define the emission probabilities of $\widehat{X}_1$ as follows:*

$$
\begin{aligned}
P[\widehat{X}_1 = \widetilde{z} \mid H_1] &= u \\
P[\widehat{X}_1 = z \mid H_1] &= (1 - u) \odot P[X_1 = z \mid H_1] \; \forall z \in \mathcal{X}
\end{aligned}
$$

In the above equations, $\widetilde{z}$ is a fake token added to the vocabulary at timestep 1. It follows that for any $i$, defining $\tau_i$ as in (C.1)

$$\tau_i(\widehat{e}(x)) = P[H_i, \widehat{X}_{-i} = (\widetilde{z}, \varnothing, x)_{-i}] \tag{C.2}$$

As a consequence, it follows that for $i > 1$ and any $x$ such that $(\widetilde{z}, \varnothing, x)_{-i} \in \mathrm{supp}(P[\widehat{X}_{-i}])$,

$$\overline{G}_i(\widehat{e}(x)) = P[\widehat{X}_i \mid \widehat{X}_{-i} = (\widetilde{z}, \varnothing, x)_{-i}] = W P[H_i \mid \widehat{X}_{-i} = (\widetilde{z}, \varnothing, x)_{-i}]$$

For any $x$ with $(\widetilde{z}, \varnothing, x)_{-i} \notin \mathrm{supp}(P[\widehat{X}_{-i}])$, $\overline{G}_i(\widehat{e}(x)) = \mathbf{0}$.

Next, the following lemma disentangles the influences of the fake token $\widetilde{z}$ and the input sequence on the posterior distribution of the hidden variable.

**Lemma C.2.** *In the setting above, there exists an invertible diagonal matrix $D$ such that for all $x$ such that $(\widetilde{z}, x) \in \mathrm{supp}(P[\widehat{X}_{-2}])$, the following equation holds:*

$$P[H_2 \mid \widehat{X}_1 = \widetilde{z}, \widehat{X}_{3:T+2} = x] = r_x D(P[\widehat{X}_1 = \widetilde{z}, H_2] \odot P[H_0 \mid X_{1:T} = x])$$

*Here $r_x > 0$ is a positive scalar.*

We now complete the proof of Theorem 3.6.

*Proof of Theorem 3.6.* Let $\mathcal{B}$ be the set defined in Assumption 3.5 and define $u$ such that $u_h = 1$ if $h \in \mathcal{B}$ and $u_h = 0$ otherwise. First, we restrict our focus to $x$ such that $(\widetilde{z}, x) \in \mathrm{supp}(P[\widehat{X}_{-2}])$. For these $x$, we can apply Lemma C.1 and Lemma C.2 in the manner described in the proof sketch. This gives $\overline{G}_2(\widehat{e}(x)) = r_x W D v$ for $v \triangleq (A(u \odot P[H_1])) \odot P[H_0 \mid X_{1:T} = x]$. By definition of $\mathcal{B}$, we have $\mathrm{supp}(A(u \odot P[H_1])) = \mathcal{H}^\star$, so $\mathrm{supp}(Dv) \subseteq \mathcal{H}^\star$. Thus, there is a matrix $\widehat{W^\dagger}$ such that

$$\widehat{W^\dagger} \overline{G}_2(\widehat{e}(x)) = r_x \widehat{W^\dagger} W D v = r_x W D v$$

The existence of $\widehat{W^\dagger}$ is due to the fact that $\{W_{:,h}\}_{h \in \mathcal{H}^\star}$ is a linearly independent set of vectors, and $\mathrm{supp}(Dv) \subseteq \mathcal{H}^\star$ whenever $x$ satisfies $(\widetilde{z}, x) \in \mathrm{supp}(P[\widehat{X}_{-2}])$. Next, we note that a matrix $B$ exists such that $(BDv)_h = \Pr(H_0 = h \mid X_{1:T} = x)$ for $h \in \mathcal{H}^\star$ and $(BDv)_h = 0$ otherwise. This is because $D$ is invertible, and $\mathrm{supp}(A(u \odot P[H_1])) = \mathcal{H}^\star$, so we can recover $P[H_0 \mid X_{1:T} = x]$ on coordinates in $\mathcal{H}^\star$ by applying another coordinate-wise scaling. It follows that we can set $b = \mu^\top B \widehat{W^\dagger}$. With this choice of $b$, we compute

$$b^\top \overline{G}_2(\widehat{e}(x)) = r_x \mu^\top B D v = r_x \sum_{h \in \mathcal{H}^\star} \mu_h \Pr(H_0 = h \mid X_{1:T} = x) = r_x \mu^\top P[H_0 \mid X_{1:T} = x]$$

where the last equality follows because $\mathrm{supp}(\mu) \subseteq \mathcal{H}^\star$. This completes the case where $(\widetilde{z}, x) \in \mathrm{supp}(P[\widehat{X}_{-2}])$.

Otherwise, for $(\widetilde{z}, x) \notin \mathrm{supp}(P[\widehat{X}_{-2}])$, by the behavior of $\overline{G}$ in Lemma C.1, $\overline{G}_2(\widehat{e}(x)) = \mathbf{0}$, so any linear head must output $b^\top \overline{G}_2(\widehat{e}(x)) = \mathbf{0}$. Furthermore, by the conditional independence structure in $\widehat{X}$, we must also have $\mathrm{supp}(P[H_2, \widehat{X}_1 = \widetilde{z}]) \cap \mathrm{supp}(P[H_2, \widehat{X}_{3:T+2} = x]) = \varnothing$. As $\mathrm{supp}(\mu) \subseteq \mathrm{supp}(P[H_2, \widehat{X}_1 = \widetilde{z}])$, this must also mean $\mathrm{supp}(\mu) \cap \mathrm{supp}(P[H_2, \widehat{X}_{3:T+2} = x]) = \varnothing$. However, we also have $P[H_2, \widehat{X}_{3:T+2} = x] = P[H_2, X_{3:T+2} = x]$ by the definition of $\widehat{X}$, and this must have the same support as $P[H_0 \mid X_{1:T} = x]$ by applying Claim B.1 and the fact that $x \in \mathrm{supp}(P[X])$. It follows that for this choice of $x$, $\mu^\top P[H_0 \mid X_{1:T} = x] = 0$, so the desired statement still stands. $\square$

We fill in the proofs of the lemmas below.

*Proof of Lemma C.1.* First, we note that (C.2) follows directly from the derivation of $\tau$, and well-known results about message passing [16]. Next, it suffices to consider the case where $(\widetilde{z}, \varnothing, x)_{-i} \notin \mathrm{supp}(P[\widehat{X}_{-i}])$, as the other case follows directly from the definition of $\overline{G}$ in terms of $\tau$. In this case, we observe that $\tau_i(\widehat{e}(x)) = P[H_i, \widehat{X}_{-i} = (\widetilde{z}, \varnothing, x)_{-i}] = \mathbf{0}$. It follows that $\|\tau_i(\widehat{e}(x))\|_1 = 0$. Thus, from our definition of $\overline{G}$, we must have $\overline{G}_i(\widehat{e}(x)) = \mathbf{0}$. $\square$

*Proof of Lemma C.2.* By the conditional independence relations in a HMM, $\widehat{X}_1 \perp \widehat{X}_{3:T+2} \mid H_2$. Using Bayes' rule, we obtain

$$P[H_2 \mid \widehat{X}_1 = \widetilde{z}, \widehat{X}_{3:T+2} = x] = \frac{P[\widehat{X}_1 = \widetilde{z}, \widehat{X}_{3:T+2} = x \mid H_2] \odot P[H_2]}{\Pr(\widehat{X}_1 = \widetilde{z}, \widehat{X}_{3:T+2} = x)}$$

$$= \frac{P[\widehat{X}_1 = \widetilde{z} \mid H_2] \odot P[\widehat{X}_{3:T+2} = x \mid H_2] \odot P[H_2]}{\Pr(\widehat{X}_1 = \widetilde{z}, \widehat{X}_{3:T+2} = x)}$$

(by conditional independence)

$$= \frac{P[\widehat{X}_1 = \widetilde{z} \mid H_2] \odot P[X_{1:T} = x \mid H_0] \odot P[H_2]}{\Pr(\widehat{X}_1 = \widetilde{z}, \widehat{X}_{3:T+2} = x)}$$

(by definition of $\widehat{X}$ and the Markovian property)

$$= r_x P[\widehat{X}_1 = \widetilde{z}, H_2] \odot P[H_0 \mid X_{1:T} = x] \odot \frac{1}{P[H_0]}$$

Where we define $r_x \triangleq \frac{\Pr(X_{1:T} = x)}{\Pr(\widehat{X}_1 = \widetilde{z}, \widehat{X}_{3:T+2} = x)}$. We note that $r_x$ is positive and well-defined by the conditions of the lemma and Theorem 3.6. We can set $D$ to be the matrix $\mathrm{diag}(\frac{1}{P[H_0]})$, which has finite positive entries on the diagonal by Assumption 3.2. $\qquad\square$

# D   Proofs for Section 4

First, we introduce a proposition which is generally useful for proving the theorems in Section 4.

**Proposition D.1.** *In the setting of Section 4, it holds that*

$$P[X_i \mid X_{-i} = x_{-i}] = P[X_i \mid M_{J_i}, J_i, S_i]P[M_{J_i}, J_i, S_i, X_{-i} = x_{-i}]$$

*Equivalently, we have the expansion*

$$P[X_i \mid X_{-i} = x_{-i}] = \sum_{h=(j,s)} \sum_m W_{:,(m,j,s)} \Pr(M_j = m, H_i = h \mid X_{-i} = x_{-i}) \qquad (D.1)$$

*Proof.* An alternative interpretation of this statement is that $X_i$ is conditionally independent from everything else given $M_{J_i}, J_i, S_i$. However, we will prove this statement algebraically. We compute

$$P[X_i \mid X_{-i} = x_{-i}] =$$

$$\sum_{h=(j,s)} \sum_{m_j} \sum_{m_{-j}} P[X_i \mid M_{-j} = m_{-j}, M_j = m_j, H_i = h] \Pr(M_{-j} = m_{-j}, M_j = m_j, H_i = h \mid X_{-i} = x_{-i})$$

$$= \sum_{h=(j,s)} \sum_{m_j} \sum_{m_{-j}} W_{:,(m_j,j,s)} \Pr(M_{-j} = m_{-j}, M_j = m_j, H_i = h \mid X_{-i} = x_{-i})$$

$$= \sum_{h=(j,s)} \sum_{m_j} W_{:,(m_j,j,s)} \Pr(M_j = m_j, H_i = h \mid X_{-i} = x_{-i})$$

$\qquad\square$

## D.1   Proof of Theorem 4.3

Throughout this section, we use $M_{J_i}$ to denote the random variable obtained by indexing $M$ by $J_i$, both of which are themselves random variables. Let $\widehat{\mathcal{I}}$ denote the set of indices $i$ where $\mathrm{supp}(P[J_i \mid X_{-i} = x_{-i}]) = \{j^\star\}$ and $\mathrm{supp}(P[S_i \mid X_{-i} = x_{-i}]) \subseteq \mathcal{S}^\star$. We will first construct the key function $K$ and query $q$ such that the set of $\mathcal{I}$ of attended-to positions (4.2) is precisely $\widehat{\mathcal{I}}$. This construction does not require the position embeddings $\beta_1, \ldots, \beta_t$, so we set them to $\mathbf{0}$.

The following lemma demonstrates the existence of $K$ and $q$ such that $\mathcal{I} = \widehat{\mathcal{I}}$.

**Lemma D.2.** *In the setting of Theorem 4.3, define $\widehat{\mathcal{I}} \triangleq \{i : \mathrm{supp}(P[J_i \mid X_{-i} = x_{-i}]) = \{j^\star\}$ and $\mathrm{supp}(P[S_i \mid X_{-i} = x_{-i}]) \subseteq \mathcal{S}^\star\}$. Then there exist query $q \in \mathbb{R}^{|\mathcal{H}|}$ and key $K$ parameterized by $\Theta^{(K)} \in \mathbb{R}^{|\mathcal{H}| \times |\mathcal{X}|}$, such that when $x \in \mathrm{supp}(P[X])$ and $\widehat{\mathcal{I}}$ is nonempty, the set $\mathcal{I}$ of attended-to positions satisfies $\mathcal{I} = \widehat{\mathcal{I}}$.*

The proof of Lemma D.2 requires the following claim.

**Claim D.3.** *In the setting of Theorem 4.3, there is a matrix $\Theta^{(1)} \in \mathbb{R}^{|\mathcal{H}| \times |\mathcal{X}|}$ such that for all $x \in \text{supp}(P[X])$ and $s \in \mathcal{S}^\star$, $(\Theta^{(1)} G_i(x))_{(j^\star, s)} = P[H_i = (j^\star, s) \,|\, X_{-i} = x_{-i}]$. Furthermore, $\|\Theta^{(1)} G_i(x)\|_1 = 1$. In addition, for $s \in \mathcal{S}^\star$, there exists $\Theta^{(2,s)} \in \mathbb{R}^{|\mathcal{M}| \times |\mathcal{X}|}$ such that for all $x \in \text{supp}(P[X])$,*

$$\Theta^{(2,s)} G_i(x) = P[M_{j^\star}, H_i = (j^\star, s) \,|\, X_{-i} = x_{-i}]$$

*Proof.* We have, by Proposition D.1,

$$
\begin{aligned}
G_i(x) &= P[X_i \,|\, X_{-i} = x_{-i}] \\
&= \sum_{h=(j,s)} \left( \sum_m W_{:,(m,j,s)} \text{Pr}(M_j = m, H_i = h \,|\, X_{-i} = x_{-i}) \right) \\
&= \sum_{h=(j,s)} \nu^{(h)}
\end{aligned}
$$

In the last equality, we defined $\nu^{(h)}$ to be the expression in the parentheses. Note that $\nu^{(h)} \in \mathcal{V}^{(h)} \triangleq \text{span}(\{W_{:,(m,h)}\}_{m \in \mathcal{M}})$. Furthermore, for $h \notin \mathcal{H}^\star$, $\nu^{(h)} \in \bar{\mathcal{V}} \triangleq \text{span}(\{W_{:,(m,h)}\}_{m \in \mathcal{M}, h \in \mathcal{H} \setminus \mathcal{H}^\star})$. As the spans $(\mathcal{V}^{(h)})_{h \in \mathcal{H}^\star}$ and $\bar{\mathcal{V}}$ are all pairwise disjoint, by Assumption 4.2, for each $h \in \mathcal{H}^\star$, we can recover

$$\nu^{(h)} = B^{(h)} P[X_i \,|\, X_{-i} = x_{-i}]$$

Likewise, we can obtain

$$\sum_{h \notin \mathcal{H}^\star} \nu^{(h)} = \bar{B} P[X_i \,|\, X_{-i} = x_{-i}]$$

Now we have, for $h \in \mathcal{H}^\star$,

$$
\begin{aligned}
\mathbf{1}^\top \nu^{(h)} &= \sum_m \mathbf{1}^\top W_{:,(m,h)} \text{Pr}(M_j = m, H_i = h \,|\, X_{-i} = x_{-i}) \\
&= \sum_m \text{Pr}(M_j = m, H_i = h \,|\, X_{-i} = x_{-i}) \qquad \text{(because } \mathbf{1}^\top W_{:,(m,h)} = 1\text{)} \\
&= \text{Pr}(H_i = h \,|\, X_{-i} = x_{-i})
\end{aligned}
$$

Likewise, the same reasoning gives $\mathbf{1}^\top \sum_{h \notin \mathcal{H}^\star} \nu^{(h)} = \sum_{h \notin \mathcal{H}^\star} \text{Pr}(H_i = h \,|\, X_{-i} = x_{-i})$. Thus, we can choose $\Theta^{(1)}$ to be the matrix with rows $\Theta^{(1)}_{h,:} = \mathbf{1}^\top B^{(h)}$ when $h \in \mathcal{H}^\star$, and for some arbitrary $\bar{h} \notin \mathcal{H}^\star$, $\Theta^{(1)}_{\bar{h},:} = \mathbf{1}^\top \bar{B}$. We set all other rows to $\mathbf{0}$, and we can check that this satisfies the lemma requirements.

We now construct $\Theta^{(2,h)}$. We can express $\nu^{(h)}$ in a vectorized manner by writing

$$\nu^{(h)} = W_{:,(\mathcal{M},h)} P[M_j, H_i = h \,|\, X_{-i} = x_{-i}]$$

where $W_{:,(\mathcal{M},h)} \in \mathbb{R}^{|\mathcal{X}| \times |\mathcal{M}|}$ has columns $\{W_{:,(m,h)}\}_{m \in \mathcal{M}}$. Note that for $j = j^\star$, $s \in \mathcal{S}^\star$, the non-degeneracy assumptions imply that $W_{:,(\mathcal{M},j^\star,s)}$ has left inverse $W^\dagger_{:,(\mathcal{M},j^\star,s)}$. Thus, we set $\Theta^{(2,s)} = W^\dagger_{:,(\mathcal{M},j^\star,s)} B^{(j^\star,s)}$ to obtain for $s \in \mathcal{S}^\star$,

$$
\begin{aligned}
\Theta^{(2,s)} G_i(x) &= W^\dagger_{:,(\mathcal{M},j^\star,s)} B^{(j^\star,s)} P[X_i \,|\, X_{-i} = x_{-i}] \\
&= W^\dagger_{:,(\mathcal{M},j^\star,s)} W_{:,(\mathcal{M},j^\star,s)} P[M_{j^\star}, H_i = (j^\star, s) \,|\, X_{-i} = x_{-i}] \\
&= P[M_{j^\star}, H_i = (j^\star, s) \,|\, X_{-i} = x_{-i}]
\end{aligned}
$$

This gives the desired result. $\square$

*Proof of Lemma D.2.* We choose the first $|\mathcal{H}|$ entries of $q$ such that $q_h = 1$ if $h = (j^\star, s)$ for $s \in \mathcal{S}^\star$, and $q_h = 0$ otherwise. The last entry is 0. Next, we choose $\Theta^{(K)}$ so that the first $|\mathcal{H}|$ rows are $\Theta^{(1)}$, and the last row is all zeros. where $\Theta^{(1)}$ is defined in Claim D.3. With this choice of $\Theta^{(K)}$, $K(G_i(x))_h = \Pr(H_i = h | X_{-i} = x_{-i})$ for $h \in \mathcal{H}^\star$. Furthermore, $\|K(G_i(x))\|_1 = 1$, by Claim D.3.

Now we note that for all $i$, $1 = \|K(G_i(x))\|_1 \geqslant q^\top K(G_i(x))$, and for $i \in \widehat{\mathcal{I}}$, $q^\top K(G_i(x)) = \sum_{s \in \mathcal{S}^\star} \Pr(H_i = (j^\star, s) | X_{-i} = x_{-i}) = 1$ by definition of $q$ and $\widehat{\mathcal{I}}$. This implies that positions $i \in \widehat{\mathcal{I}}$ do indeed achieve the maximum attention scores. $\qquad\square$

Next, we also require a construction of the value function such that it computes the correct prediction for all $i \in \widehat{\mathcal{I}}$.

**Lemma D.4.** *In the setting of Theorem 4.3, let $\widehat{\mathcal{I}}$ be defined as in Lemma D.2. We can choose the parameters of the value function $V$, $\Theta^{(V)} \in \mathbb{R}^{|\mathcal{M}||\mathcal{H}| \times |\mathcal{X}|}$, $b \in \mathbb{R}^{|\mathcal{M}||\mathcal{H}|}$, such that when $x \in \text{supp}(P[X])$ and $\widehat{\mathcal{I}}$ is nonempty, for all $i \in \widehat{\mathcal{I}}$,*

$$V(G_i(x), e(x_i)) = r_{x,i} \mu^\top P[M_{j^\star} | X_{1:T} = x]$$

*where $r_{x,i} > 0$ is a positive scalar.*

*Proof.* We first choose $\Theta^{(V)}$ such that the rows satisfy $\Theta^{(V)}_{(m,j^\star,s),:} = \Theta^{(2,s)}_{m,:}$ when $s \in \mathcal{S}^\star$ for $\Theta^{(2,s)}$ constructed in Claim D.3, and $\Theta^{(V)}_{(m,j,s),:} = \mathbf{0}_{|\mathcal{X}|}$ otherwise for $j \neq j^\star$ or $s \notin \mathcal{S}^\star$.

We claim that for $i \in \widehat{\mathcal{I}}$,

$$\Theta^{(V)} G_i(x) = P[M_{J_i}, J_i, S_i | X_{-i} = x_{-i}] \tag{D.2}$$

This is because for $s \in \mathcal{S}^\star$, $\Theta^{(2,s)} G_i(x) = P[M_{j^\star}, H_i = (j^\star, s) | X_{-i} = x_{-i}]$ by Claim D.3, and for $h = (j, s)$ for $j \neq j^\star$ or $s \notin \mathcal{S}^\star$,

$$P[M_j, H_i = h | X_{-i} = x_{-i}] = P[M_j | H_i = h, X_{-i} = x_{-i}] \Pr(H_i = h | X_{-i} = x_{-i}) = \mathbf{0}_{|\mathcal{M}|}$$

Note that this last equality followed because $\Pr(H_i = h | X_{-i} = x_{-i}) = 0$ for the choice of $h$ and $i \in \widehat{\mathcal{I}}$. By construction of $\Theta^{(V)}$, these computations imply that (D.2) does indeed hold. The embedding can be chosen such that $e(x_i) = P[X_i = x_i | M_{J_i}, J_i, S_i]$. Thus, we have for $i \in \widehat{I}$:

$$\begin{aligned}
(\Theta^{(V)} G_i(x)) \odot e(x_i) &= P[M_{J_i}, J_i, S_i | X_{-i} = x_{-i}] \odot P[X_i = x_i | M_{J_i}, J_i, S_i] \\
&= P[X_i = x_i, M_{J_i}, J_i, S_i | X_{-i} = x_{-i}]
\end{aligned}$$

The last equality followed from applying the same reasoning as in Proposition D.1.

Now we let $B \in \mathbb{R}^{|\mathcal{M}| \times |\mathcal{M}||\mathcal{H}|}$ be the matrix such that

$$(BP[X_i = x_i, M_{J_i}, (J_i, H_i) | X_{-i} = x_{-i}])_m =$$
$$\sum_s \Pr(X_i = x_i, M_{j^\star} = m, J_i = j^\star, S_i = s | X_{-i} = x_{-i})$$

Now we pick the last linear weight in the value function by $b = B^\top \mu$. It follows that for $i \in \widehat{\mathcal{I}}$,

$$\begin{aligned}
V(G_i(x), e(x_i)) &= b^\top ((\Theta^{(V)} G_i(x)) \odot e(x_i)) \\
&= \mu^\top B((\Theta^{(V)} G_i(x)) \odot e(x_i)) \\
&= \mu^\top B P[X_i = x_i, M_{J_i}, J_i, S_i | X_{-i} = x_{-i}] \\
&= \mu^\top \sum_s P[X_i = x_i, M_{j^\star}, J_i = j^\star, S_i = s | X_{-i} = x_{-i}] \\
&= \mu^\top P[M_{j^\star}, X_i = x_i | X_{-i} = x_{-i}]
\end{aligned}$$

We obtained the last equality by observing that $\sum_s P[X_i = x_i, M_{j^\star}, J_i = j^\star, S_i = s | X_{-i} = x_{-i}] = P[M_{j^\star}, X_i = x_i | X_{-i} = x_{-i}]$ for $i \in \widehat{\mathcal{I}}$, as the distribution of $H_i$ must concentrate where $J_i = j^\star$. Finally, we observe that $\mu^\top P[M_{j^\star}, X_i = x_i | X_{-i} = x_{-i}] = \mu^\top P[M_{j^\star} | X_{1:T} = x] \Pr(X_i = x_i | X_{-i} = x_{-i})$, so setting $r_{x,i} = \Pr(X_i = x_i | X_{-i} = x_{-i})$ completes the proof. $\qquad\square$

Now we can complete the proof of Theorem 4.3.

*Proof of Theorem 4.3.* By applying Lemmas D.2 and D.4, we constructed key, query, and value functions for the attention head such that for all $x \in \mathrm{supp}(P[X])$ with $\widehat{\mathcal{I}}$ (defined in Lemma D.2) nonempty, the attended-to positions $\mathcal{I}$ satisfy $\mathcal{I} = \widehat{\mathcal{I}}$, and $V(G_i(x), e(x_i)) = r_{x,i}\mu^\top P[M_{j^\star} \mid X_{1:T} = x]$. As the attention head computes the average of $V(G_i(x), e(x_i))$ over attended-to positions, and $r_{x,i}$ is positive for all $i \in \widehat{\mathcal{I}}$, we obtain the desired result. $\square$

We note that this proof also works for the case where there is a single memory cell, as that is a special case where $J_i = j^\star$ always, and we only need to consider the evolution of $S_i$.

### D.2 Formal abstraction for prompt tuning in Section 4.2

We will work directly in the case with multiple memories, as the single memory case is captured in this setting. We follow the construction in Section C. our message passing formulation requires the augmented Markov chain $\widetilde{H}_0 \triangleq (M_1, \ldots, M_N, H_0)$, $\widetilde{H}_1 \triangleq (M_1, \ldots, M_N, H_1)$, ..., which uses the following transition probabilities:

$$\Pr(\widetilde{H}_{i+1} = (m', h') \mid \widetilde{H}_i = (m, h)) = A_{h',h}\mathbb{1}(m' = m)$$

Let $\widetilde{\mathcal{H}}$ denote the set of possible values for $\widetilde{H}$. For vector $v \in \mathbb{R}^{|\mathcal{M}||\mathcal{H}|}$ we define a lifting function $\eta : \mathbb{R}^{|\mathcal{M}||\mathcal{H}|} \to \mathbb{R}^{|\widetilde{\mathcal{H}}|}$ by

$$\eta(v)_{(m_{1:N}, j, s)} = v_{(m_j, j, s)}$$

We observe that $\eta(P[X_i = x_i \mid M_{J_i}, (J_i, S_i)]) = P[X_i = x_i \mid \widetilde{H}_i]$.

Now we formalize the model $\overline{G}$. $\overline{G}$ will take embedding vectors $v = (v_1, \ldots, v_t)$ with $v_i \in \mathbb{R}^{|\widetilde{\mathcal{H}}|}$ as follows. We define left and right messages $\overleftarrow{\delta}_{i+1 \to i}(v)$ and $\overrightarrow{\delta}_{i-1 \to i}(v)$ for $i \in [t]$ via:

$$\overleftarrow{\delta}_{t+1 \to t}(v) = P[\widetilde{H}_t]$$
$$\overleftarrow{\delta}_{i \to i-1}(v) = P[\widetilde{H}_{i-1} \mid \widetilde{H}_i](\overleftarrow{\delta}_{i+1 \to i}(v) \odot v_i) \; \forall 1 < i < t$$
$$\overrightarrow{\delta}_{0 \to 1}(v) = P[\widetilde{H}_1]$$
$$\overrightarrow{\delta}_{i \to i+1}(v) = P[\widetilde{H}_{i+1} \mid \widetilde{H}_i](\overrightarrow{\delta}_{i-1 \to i}(v) \odot v_i) \; \forall 1 < i < t$$

We observe that this definition almost matches Section C, except it replaces $H$ with $\widetilde{H}$. Next, we define the aggregated message at timestep $i$ by

$$\tau_i(v) = \begin{cases} \overleftarrow{\delta}_{2 \to 1}(v) & \text{if } i = 1 \\ \dfrac{\overleftarrow{\delta}_{i+1 \to i}(v) \odot \overrightarrow{\delta}_{i-1 \to i}(v)}{P[\widetilde{H}_i]} & \text{if } 1 < i < t \\ \overrightarrow{\delta}_{t-1 \to t}(v) & \text{if } i = t \end{cases} \tag{D.3}$$

In the edge case where $P[M]$ does not have full support, the coordinate-wise division in the definition above would sometimes divide by 0. However, for all these cases both of the corresponding terms in the numerator must also be 0, so we can simply set the value of $\tau_i$ in this coordinate to 0. We will see that this preserves the meaning of the message $\tau_i$, which for the proper embeddings $e(x_i) = P[X_i = x_i \mid \widetilde{H}_i]$, with $e(x) = (e(x_1), \ldots, e(x_t))$, computes

$$\tau_i(e(x)) = P[\widetilde{H}_i, X_{-i} = x_{-i}]$$

We can now define the reverse lifting function $\phi : \mathbb{R}^{|\widetilde{\mathcal{H}}| \to |\mathcal{M}||\mathcal{H}|}$ as follows:

$$(\phi(v))_{m_j, j, s} = \frac{1}{|\mathcal{M}|^{N-1}} \sum_{m_{-j}} v_{m_{1:N}, j, s} \tag{D.4}$$

We observe that $\phi(\tau_i(e(x))) = \frac{P[M_{J_i}, J_i, S_i, X_{-i} = x_{-i}]}{|\mathcal{M}|^{N-1}}$. We now compute the model output as follows:

$$\overline{G}_i(v) = W \frac{\phi(\tau_i(v))}{\|\phi(\tau_i(v))\|_1}$$

In the edge case where $\|\phi(\tau_i(v))\|_1 = 0$, we again define $\overline{G}(v) = \mathbf{0}_{|\mathcal{X}|}$. We can observe that $\overline{G}_i(e(x)) = P[X_i \mid X_{-i} = x_{-i}]$.

The downstream classifier uses the embedding $\widehat{e}(x)$ defined as follows:

$$\widehat{e}(x) = (u, e(x_1)), \ldots, e(x_t)))$$

with a tunable prompt embedding $u \in \mathbb{R}^{|\widetilde{\mathcal{H}}|}$. We also require a slightly modified attention head. The value function $V$ in the attention head is slightly modified to accomodate the new embedding dimension. Letting $V : \mathbb{R}^{|\mathcal{X}|} \times \mathbb{R}^{|\widetilde{\mathcal{H}}|} \to \mathbb{R}$,

$$V(a, v) = b^\top((\Theta^{(V)}a) \odot \phi(v))$$

The dimensions of the parameters $b, \Theta^{(V)}$ remain unchanged. Note that when there is just a single memory, this reduces to the case in Section 4.

## D.3 Analysis for prompt tuning in the multiple memory setting

We will state and prove our result for the prompt tuning setting with multiple memories. For the multiple memory setting, the downstream classifier uses the following embedding function $\widehat{e}$:

$$\widehat{e}(x) = (u, \eta(e(x_1)), \ldots, \eta(e(x_t)))$$

with a tunable prompt embedding $u \in \mathbb{R}^{|\widetilde{\mathcal{H}}|}$. The attention head is changed so that the value function takes a larger dimensional embedding:

$$V(a, v) = b^\top((\Theta^{(V)}a) \odot \phi(v))$$

where $\phi$ is defined in (D.4). The following assumption extends Assumption 4.5 to the multiple memory case.

**Assumption D.5** (Multiple memories version of Assumption 4.5). *Let $\mathcal{M}^\star \triangleq \mathrm{supp}(\mu)$ denote the set of non-zero coordinates in $\mu$. There exists a set of recoverable hidden states $\mathcal{H}^\star$, such that the collection of token emission probabilities from $\mathcal{M}^\star \times \mathcal{H}^\star$, $\{W_{:,(m,h)}\}_{m \in \mathcal{M}^\star, h \in \mathcal{H}^\star}$, is a linearly independent set of vectors.*

*Furthermore, define the following span of vectors:*

$$\overline{\mathcal{V}} \triangleq \mathrm{span}(\{W_{:,(m,j^\star,s)}\}_{m \in \mathcal{M}^\star, s \in \mathcal{S} \setminus \mathcal{S}^\star} \cup \{W_{:,(m,j,s)}\}_{m \in \mathcal{M}, j \neq j^\star, s \in \mathcal{S}})$$

*Then $\overline{\mathcal{V}}$ must be disjoint from the span of token emission probabilities from $\mathcal{M}^\star \times \mathcal{H}^\star$:*

$$\mathrm{span}(\{W_{:,(m,h)}\}_{m \in \mathcal{M}^\star, h \in \mathcal{H}^\star}) \cap \overline{\mathcal{V}} = \{\mathbf{0}_{|\mathcal{X}|}\}$$

Note that Assumption D.5 reduces to Assumption 4.5 the case where $N$, the number of memory cells, is 1. In any case, it is a relaxation of Assumption 4.2.

We now state and prove the result for multiple memories.

**Theorem D.6.** *In the setting above, suppose that non-degeneracy Assumption D.5 and holds. In addition, suppose that Assumption 4.4 (stationarity) holds. Then there exists a prompt $u$ and attention head on $\overline{G}(\widehat{e}(x))$ and the token embeddings which can compute the ground-truth $F^\star(x)$ for any $x \in \mathcal{R}$, defined in (4.3):*

$$F^\star(x) = \mathbb{1}(\mathrm{Attn}((\overline{G}_i(\widehat{e}(x)), \widehat{e}_i(x))_{i=1}^{t+1}) \geqslant 0)$$

*Here $\widehat{e}$ is the embedding in (4.4) and $\mathrm{Attn}$ is defined in (4.2).*

We begin by rigorously stating the observation that soft prompt tuning is equivalent to adding a fake token $\widetilde{z}$ to the vocabulary and modifying the token emission probabilities at timestep 1, analogous to Lemma C.1.

**Lemma D.7.** *In the setting of Theorem D.6, define $\widetilde{H}$ as in Section D.2. Fix any prompt vector $u \in [0, 1]^{|\widetilde{\mathcal{H}}|}$. Define the random variable $\widehat{X}$ with the same emission probabilities as $X$ for $i > 1$: $P[\widehat{X}_i \mid \widetilde{H}_i] = P[X_i \mid \widetilde{H}_i]$. For timestep 1, we define the emission probabilities of $\widehat{X}_1$ as follows:*

$$P[\widehat{X}_1 = \widetilde{z} \mid \widetilde{H}_1] = u$$
$$P[\widehat{X}_1 = z \mid \widetilde{H}_1] = (1 - u) \odot P[X_1 = z \mid \widetilde{H}_1] \; \forall z \in \mathcal{X}$$

In the above equations, $\widetilde{z}$ is a fake token added to the vocabulary at timestep 1. It follows that for any $i$, defining $\tau_i$ as in (D.3)

$$\tau_i(\widehat{e}(x)) = P[\widetilde{H}_i, \widehat{X}_{-i} = (\widetilde{z}, x)_{-i}] \tag{D.5}$$

As a consequence, it follows that for $i > 1$ and any $x$ such that $(\widetilde{z}, x)_{-i} \in \mathrm{supp}(P[\widehat{X}_{-i}])$,

$$\overline{G}_i(\widehat{e}(x)) = P[\widehat{X}_i \mid \widehat{X}_{-i} = (\widetilde{z}, x)_{-i}] = W P[M_{J_i}, J_i, S_i \mid \widehat{X}_{-i} = (\widetilde{z}, x)_{-i}]$$

For any $i$ and $x$ with $(\widetilde{z}, x)_{-i} \notin \mathrm{supp}(P[\widehat{X}_{-i}])$, $\overline{G}_i(\widehat{e}(x)) = \mathbf{0}$.

The proof of Lemma D.7 mirrors the proof of Lemma C.1, so we omit it here.

In particular, throughout the proof we will use the following prompt $u$:

$$u_{m_{1:N}, j, s} = \begin{cases} 1 & \text{if } m_{j^\star} \in \mathrm{supp}(\mu) \\ 0 & \text{otherwise} \end{cases} \tag{D.6}$$

We will also use the notation $\widehat{x} \triangleq (\widetilde{z}, x_1, \dots, x_t)$. The following lemma considers behaviors in edge cases with this choice of $u$.

Towards our proofs, the following result is useful.

**Proposition D.8.** *In the setting of Theorem D.6, where $P[H_0]$ is the stationary distributions satisfying $P[H_0] = AP[H_0]$, it holds that*

$$P[M, H_i, X_{i+1:i+t}] = P[M, H_0, X_{1:t}]$$

*for any $t \geqslant 1$, $i \geqslant 1$.*

*Proof.* Because $P[H_0]$ is stationary, we observe that $P[M, H_i] = P[M, H_0]$ for all $i$. We write

$$
\begin{aligned}
P[X_{i+1:i+t}, M = m, H_i = h] &= P[X_{i+1:i+t} \mid M = m, H_i = h]\mathrm{Pr}(M = m, H_i = h) \\
&= P[X_{1:t} \mid M = m, H_0 = h]\mathrm{Pr}(M = m, H_i = h) \\
&\qquad\qquad\qquad\qquad\text{(by time-invariance of HMMs)} \\
&= P[X_{1:t} \mid M = m, H_0 = h]\mathrm{Pr}(M = m, H_0 = h)
\end{aligned}
$$

$\square$

We will now restrict our focus to the set of inputs

$$\mathcal{Z} \triangleq \{x : \mathrm{Pr}(\widehat{X}_{-i} = (\widetilde{z}, x)_{-i}) > 0 \; \forall i \in [t]\} \tag{D.7}$$

We also define the set

$$\widehat{\mathcal{I}} \triangleq \{i + 1 : \mathrm{supp}(P[S_i | X_{-i} = x_{-i}]) \subseteq \mathcal{S}^\star, \mathrm{supp}(P[J_i | X_{-i} = x_{-i}]) \subseteq \{j^\star\}, i \in [t]\} \tag{D.8}$$

Here $\mathcal{S}^\star$ is defined in the non-degeneracy assumption. We will first construct key and query parameters such that the set of attended-to positions is precisely $\widehat{\mathcal{I}}$, following the proof of Theorem 4.3.

**Lemma D.9** (Analogue to Lemma D.2)**.** *In the setting of Theorem D.6 and above, define $u$ as in (D.6). There are parameters $\Theta^{(K)} \in \mathbb{R}^{(|\mathcal{H}|+1) \times |\mathcal{X}|}$, $q \in \mathbb{R}^{|\mathcal{H}|+1}$, and $\beta_1, \beta_2, \dots \in \mathbb{R}^{|\mathcal{H}|+1}$ such that for any $x \in \mathcal{Z}$ where $\widehat{\mathcal{I}}$ is nonempty, the set of attended-to positions $\mathcal{I}$ (defined in (4.1)) satisfies $\mathcal{I} = \widehat{\mathcal{I}}$.*

Towards proving Lemma D.9, the following construction will be useful.

**Claim D.10** (Analogue of Claim D.3)**.** *In the setting of Theorem D.6, define $\mathcal{H}^\star$ as in Assumption D.5. There is a matrix $\Theta^{(1)} \in \mathbb{R}^{|\mathcal{H}| \times |\mathcal{X}|}$ such that for all $x \in \mathrm{supp}(P[X])$, and $i > 1$ with $\mathrm{Pr}(\widehat{X}_{-i} = \widehat{x}_{-i}) > 0$, $(\Theta^{(1)} \overline{G}_i(\widehat{e}(x)))_h = P[H_i = h \mid \widehat{X}_{-i} = \widehat{x}_{-i}]$ for any $h \in \mathcal{H}^\star$. Furthermore, $\|\Theta^{(1)} \overline{G}_i(\widehat{e}(x))\|_1 = 1$.*

*In addition, for $s \in \mathcal{S}^\star$, there exists $\Theta^{(2,s)} \in \mathbb{R}^{|\mathcal{M}| \times |\mathcal{X}|}$ such that for all $i > 1$ and $x$ with $\mathrm{Pr}(\widehat{X}_{-i} = \widehat{x}_{-i}) > 0$,*

$$\Theta^{(2,s)} \overline{G}_i(\widehat{e}(x)) = P[M_{j^\star}, H_i = (j^\star, s) \mid \widehat{X}_{-i} = \widehat{x}_{-i}]$$

Our proof will require the following result which shows that the distribution of $M_{j^\star}$ has limited support.

**Proposition D.11.** *In the setting of Theorem D.6 and Lemma D.7, let $u$ be defined as in* (D.6). *Then for all $i > 1$,* $\text{supp}(P[M_{j^\star} \mid \widehat{X}_{-i} = \widehat{x}_{-i}]) \subseteq \text{supp}(\mu)$ *if* $\Pr(\widehat{X}_{-i} = \widehat{x}_{-i}) > 0$.

*Proof.* We have

$$P[M_{j^\star} \mid \widehat{X}_{-i} = \widehat{x}_{-i}] = \sum_{m_{-j^\star}, h} P[M_{j^\star}, M_{-j^\star} = m_{-j^\star}, H_1 = h \mid \widehat{X}_{-i} = \widehat{x}_{-i}]$$

$$= \sum_{m_{-j^\star}, h} \frac{P[\widehat{X}_1 = \widetilde{z} \mid M_{j^\star}, M_{-j^\star} = m_{-j^\star}, H_1 = h] \odot P[M_{j^\star}, M_{-j^\star} = m_{-j^\star}, H_1 = h \mid \widehat{X}_{-(1,i)} = \widehat{x}_{-(1,i)}]}{\Pr(\widehat{X}_1 = \widetilde{z} \mid \widehat{X}_{-(1,i)} = \widehat{x}_{-(1,i)})}$$

In this equation we used $_{-(1,i)}$ to index all but the first and $i$-th element of the sequence. We note that $\text{supp}(P[\widehat{X}_1 = \widetilde{z} \mid M_{j^\star}, M_{-j^\star} = m_{-j^\star}, H_1 = h]) = \text{supp}(\mu)$ for all $m_{-j^\star}, h$, so the desired statement follows. $\square$

Now we complete the proof of Claim D.10.

*Proof of Claim D.10.* The proof of this statement will be analogous to Claim D.3. As before, we have

$$G_i(\widehat{e}(x)) = \sum_{h=(j,s)} \left( \sum_m W_{:,(m,j,s)} \Pr(M_j = m, H_i = h \mid \widehat{X}_{-i} = \widehat{x}_{-i}) \right)$$
$$= \sum_{h=(j,s)} \nu^{(h)}$$

In the last equality, we defined $\nu^{(h)}$ to be the expression in the parentheses. We consider several cases. First, when $h = (j^\star, s)$ for $s \in \mathcal{S}$, we must have that when $i > 1$, $P[M_{j^\star} \mid \widehat{X}_{-i} = \widehat{x}_{-i}]$ is supported on $\mathcal{M}^\star$ by Proposition D.11. Thus, $\nu^{(h)} \in \mathcal{V}^{(h)} \triangleq \text{span}(\{W_{:,(m,h)}\}_{m \in \mathcal{M}^\star})$. As a result, for $h \notin \mathcal{H}^\star$, $\nu^{(h)} \in \overline{\mathcal{V}}$, which is the span of vectors defined in Assumption D.5. As the spans $(\mathcal{V}^{(h)})_{h \in \mathcal{H}^\star}$ and $\overline{\mathcal{V}}$ are all pairwise disjoint, by Assumption 4.2, for each $h \in \mathcal{H}^\star$, we can recover

$$\nu^{(h)} = B^{(h)} P[X_i \mid X_{-i} = x_{-i}]$$

Likewise, we can obtain

$$\sum_{h \notin \mathcal{H}^\star} \nu^{(h)} = \bar{B} P[X_i \mid X_{-i} = x_{-i}]$$

The remainder of this proof for the construction of $\Theta^{(1)}$ follows the same steps as Claim D.3.

For the second part about constructing $\Theta^{(2,s)}$, we modify Claim D.3 in a few ways. First, each $\nu^{(j^\star, s)}$ is recoverable as a linear function of $\overline{G}_i(\widehat{e}(x))$ when $s \in \mathcal{S}^\star$. Now using $\mathcal{M}^\star \subseteq \mathcal{M}$ as shorthand for $\text{supp}(\mu)$, we define the matrix $W^\dagger_{:,(\mathcal{M}^\star, j^\star, s)} \in \mathbb{R}^{|\mathcal{M}^\star| \times |\mathcal{X}|}$ to be the left inverse of $W_{:,(\mathcal{M}^\star, j^\star, s)}$, the matrix with columns $\{W_{:,(m,j^\star,s)}\}_{m \in \mathcal{M}^\star}$. This left inverse exists by the non-degeneracy assumptions. Now we construct the matrix $\widetilde{W^\dagger_{:,(\mathcal{M}^\star, j^\star, s)}} \in \mathbb{R}^{|\mathcal{M}| \times |\mathcal{X}|}$, where the $m$-th row of $\widetilde{W^\dagger_{:,(\mathcal{M}^\star, j^\star, s)}}$ matches the corresponding row of $W^\dagger_{:,(\mathcal{M}^\star, j^\star, s)}$ if $m \in \mathcal{M}^\star$ and is $\mathbf{0}$ otherwise.

We observe that because $\text{supp}(P[M_{j^\star}, H_i = (j^\star, s) \mid \widehat{X}_{-i} = \widehat{x}_{-i}]) \subseteq \mathcal{M}^\star$ by Proposition D.11, we can finish the proof by repeating the argument of Claim D.3. $\square$

The following claim relating the support of $H_i$ conditioned on $\widehat{X}$ to the support of $H_i$ conditioned on $X$ will also be useful.

**Claim D.12.** *In the setting of Theorem D.6 and Lemma D.7, suppose that $u$ is defined as in* (D.6). *For $i > 1$ with $\Pr(\widehat{X}_{-i} = \widehat{x}_{-i}) > 0$, we have*

$$\text{supp}(P[H_i \mid \widehat{X}_{-i} = \widehat{x}_{-i}]) \subseteq \text{supp}(P[H_{i-1} \mid X_{-(i-1)} = x_{-(i-1)}])$$

*Proof.* We have

$$P[H_i \,|\, \widehat{X}_{-i} = \widehat{x}_{-i}] = \sum_{m,h} P[M = m, H_1 = h, H_i \,|\, \widehat{X}_{-i} = \widehat{x}_{-i}] =$$

$$\frac{\sum_{m,h} \Pr(\widehat{X}_1 = \widetilde{z} \,|\, M = m, H_1 = h) P[M = m, H_1 = h, H_i \,|\, \widehat{X}_{2:i-1} = \widehat{x}_{2:i-1}, \widehat{X}_{i+1:T+1} = \widehat{x}_{i+1:t+1}]}{\Pr(X_1 = \widetilde{z} \,|\, \widehat{X}_{2:i-1} = \widehat{x}_{2:i-1}, \widehat{X}_{i+1:T+1} = \widehat{x}_{i+1:t+1})}$$

$$= \frac{\sum_{m,h} \Pr(\widehat{X}_1 = \widetilde{z} \,|\, M = m, H_1 = h) P[M = m, H_0 = h, H_{i-1} \,|\, X_{-(i-1)} = x_{-(i-1)}]}{\Pr(X_1 = \widetilde{z} \,|\, \widehat{X}_{2:i-1} = \widehat{x}_{2:i-1}, \widehat{X}_{i+1:T+1} = \widehat{x}_{i+1:t+1})}$$
(D.9)

The last line used the time-invariance property of the HMM (Proposition D.8), the definition of $\widehat{x}$, and the fact that $P[\widehat{X}_i \,|\, H_i, M]$ is distributed the same as $P[X_i \,|\, H_i, M]$ for $i > 1$. On the other hand, note that $P[H_{i-1} \,|\, X_{-(i-1)} = x_{-(i-1)}] = \sum_{m,h} P[M = m, H_0 = h, H_{i-1} \,|\, X_{-(i-1)} = x_{-(i-1)}]$. This involves a sum over the same terms in the numerator in (D.9). Thus, as all the terms in the sum of (D.9) are nonnegative, the desired statement follows. $\square$

This lets us complete the proof of Lemma D.9.

*Proof of Lemma D.9.* By setting $\Theta^{(K)} = \begin{bmatrix} \Theta^{(1)} \\ \mathbf{0} \end{bmatrix}$, where $\Theta^{(1)}$ is defined in Claim D.10, we obtain $K$ such that for all $i > 1$, $(K(\overline{G}_i(\widehat{e}(x))))_h = \Pr(H_i = h | \widehat{X}_{-i} = \widehat{x}_{-i})$ for $h \in \mathcal{H}^\star$. Furthermore, $(K(\overline{G}_i(\widehat{e}(x))))_{|\mathcal{H}|+1} = 0$, and $\|K(\overline{G}_i(\widehat{e}(x)))\|_1 = 1$. We choose $\beta_1 = \begin{bmatrix} \mathbf{0}_{|\mathcal{H}|} \\ -2 \end{bmatrix}$ and $\beta_i = \mathbf{0}_{|\mathcal{H}|+1}$ for $i > 1$. We also construct $q$ so that the first $|\mathcal{H}|$ dimensions are the indicator on the set $\{j^\star\} \times \mathcal{S}^\star$. We set $q_{|\mathcal{H}|+1} = 1$. Note that this construction ensures that for $i > 1$, $1 = \|K(\overline{G}_i(\widehat{e}(x)))\|_1 \geqslant q^\top(K(\overline{G}_i(\widehat{e}(x))) + \beta_i) \geqslant 0$. Note that for $i \in \widehat{\mathcal{I}}$, by Claim D.12 we have $\mathrm{supp}(P[H_i \,|\, \widehat{X}_{-i} = \widehat{x}_{-i}]) \subseteq \mathrm{supp}(P[H_{i-1} \,|\, X_{-(i-1)} = x_{-(i-1)}]) \subseteq \{j^\star\} \times \mathcal{S}^\star$. Thus, for such $i \in \widehat{\mathcal{I}}$, we have $q^\top(K(\overline{G}_i(\widehat{e}(x))) + \beta_i) = 1$, achieving the maximum over all positions. Finally, we note that $1 \notin \mathcal{I}$ because the position embedding $\beta_1$ ensures that $q^\top(K(\overline{G}_1(\widehat{e}(x))) + \beta_1) \leqslant -1$. Thus, $\mathcal{I} = \widehat{\mathcal{I}}$, as desired. $\square$

Next, the following lemma constructs the value function, analogously to Lemma D.4.

**Lemma D.13** (Analogue to Lemma D.4). *In the setting of Theorem D.6 and Lemma D.7, define $u$ as in (D.6), and $\widehat{\mathcal{I}}$ as in (D.8). We can choose the parameters of the value function $V$, $\Theta^{(V)} \in \mathbb{R}^{|\mathcal{M}||\mathcal{H}| \times |\mathcal{X}|}$, $b \in \mathbb{R}^{|\mathcal{M}||\mathcal{H}|}$, such that for $x \in \mathrm{supp}(P[X])$ where $\widehat{\mathcal{I}}$ is nonempty, for all $i \in \widehat{\mathcal{I}}$ with $\Pr(\widehat{X}_{-i} = \widehat{x}_{-i}) > 0$,*

$$V(\overline{G}_i(\widehat{e}(x)), \widehat{e}_i(x)) = \mu^\top P[\widehat{X}_i = \widehat{x}_i, M_{j^\star} \,|\, \widehat{X}_{-i} = \widehat{x}_{-i}]$$

*As a consequence, for all $i \in \widehat{\mathcal{I}}$,*

$$V(\overline{G}_i(\widehat{e}(x)), \widehat{e}_i(x)) = r_{x,i} \mu^\top P[M_{j^\star} \,|\, X = x]$$

*where $r_{x,i} > 0$ is a positive scalar. In particular, this holds regardless of whether $\Pr(\widehat{X}_{-i} = \widehat{x}_{-i}) > 0$. Furthermore, when $\widehat{x} \notin \mathrm{supp}(P[\widehat{X}])$, for all $i > 1$, we must have*

$$V(\overline{G}_i(\widehat{e}(x)), \widehat{e}_i(x)) = 0$$

We rely on the following claim.

**Claim D.14.** *In the setting of Theorem D.6 and Lemma C.1 where $u$ takes the value in in (D.6), for all $x$ where $\widehat{x} \triangleq (\widetilde{z}, x) \in \mathrm{supp}(P[\widehat{X}])$, we have*

$$\mu^\top P[M_{j^\star} \,|\, \widehat{X} = \widehat{x}] = \frac{\mu^\top P[M_{j^\star} \,|\, X_{1:T} = x]}{\Pr(\widehat{X}_1 = \widetilde{z} | \widehat{X}_{2:T+1} = \widehat{x}_{2:t+1})}$$

*Proof.* We observe that

$$\mu^\top P[M \,|\, \widehat{X} = \widehat{x}]$$

(D.10)

$$= \mu^\top \sum_h \sum_{m_{-j^\star}} P[M_{j^\star}, M_{-j^\star} = m_{-j^\star}, H_1 = h \,|\, \widehat{X} = \widehat{x}]$$

$$= \mu^\top \frac{\sum_h \sum_{m_{-j^\star}} P[\widehat{X}_1 = \widetilde{z} \,|\, M_{j^\star}, M_{-j^\star} = m_{-j^\star}, H_1 = h] \odot P[M_{j^\star}, M_{-j^\star} = m_{-j^\star}, H_1 = h \,|\, \widehat{X}_{2:T+1} = \widehat{x}_{2:t+1}]}{\Pr(\widehat{X}_1 = \widetilde{z} \,|\, \widehat{X}_{2:T+1} = \widehat{x}_{2:t+1})}$$

$$= \mu^\top \frac{\sum_h \sum_{m_{-j^\star}} P[\widehat{X}_1 = \widetilde{z} \,|\, M_{j^\star}, M_{-j^\star} = m_{-j^\star}, H_1 = h] \odot P[M_{j^\star}, M_{-j^\star} = m_{-j^\star}, H_0 = h \,|\, X_{1:T} = x]}{\Pr(\widehat{X}_1 = \widetilde{z} \,|\, \widehat{X}_{2:T+1} = \widehat{x}_{2:t+1})}$$

(by Proposition D.8 and the definition of $\widehat{X}$)

Now we have $\mu^\top \mathrm{diag}(P[\widehat{X}_1 = \widetilde{z} \,|\, M_{j^\star}, M_{-j^\star} = m_{-j^\star}, H_1 = h]) = \mu^\top$ because by construction, $P[\widehat{X}_1 = \widetilde{z} \,|\, M_{j^\star}, M_{-j^\star} = m_{-j^\star}, H_1 = h]$ is only supported on $\mathrm{supp}(\mu)$ and equals 1 on the support. Thus, we obtain

$$\mu^\top P[M_{j^\star} \,|\, \widehat{X} = \widehat{x}] = \frac{\sum_h \mu^\top P[M_{j^\star}, H_0 = h \,|\, X_{1:T} = x]}{\Pr(\widehat{X}_1 = \widetilde{z} \,|\, \widehat{X}_{2:T+1} = \widehat{x}_{2:t+1})}$$

$$= \frac{\mu^\top P[M_{j^\star} | X_{1:T} = x]}{\Pr(\widehat{X}_1 = \widetilde{z} \,|\, \widehat{X}_{2:T+1} = \widehat{x}_{2:t+1})}$$

$\square$

We also require the following result to handle edge cases where probability values are 0.

**Claim D.15.** *In the setting of Theorem D.6 and Lemma D.7, define $u$ as in (D.6). Consider an input $x \in \mathrm{supp}(P[X])$ such that $\widehat{x} \triangleq (\widetilde{z}, x_1, \ldots, x_t)$ satisfies $\Pr(\widehat{X} = \widehat{x}) = 0$. Then $\mu^\top P[M_{j^\star} | X_{1:T} = x] = 0$. Furthermore, for any $x$ where $\Pr(\widehat{X}_{-i} = \widehat{x}_{-i}) = 0$ for some $i$, we must have $\overline{G}_i(\widehat{e}(x)) = \mathbf{0}_{|\mathcal{X}|}$.*

*Proof.* First, we observe that

$$0 = \Pr(\widehat{X} = \widehat{x})$$
$$= P[\widehat{X}_1 = \widetilde{z} \,|\, M, H_1]^\top P[M, H_1, \widehat{X}_{-1} = \widehat{x}_{-1}]$$
$$= u^\top P[M, H_0, X = x] \qquad \text{(by Proposition D.8 and Lemma D.7)}$$

In particular, as $\mathrm{supp}(u) \cap \mathrm{supp}(P[M, H_0, X_{1:T} = x]) = \varnothing$, it follows that $\Pr(M_{j^\star} = m, H_0 = h, X_{1:T} = x) = 0$ for all $m \in \mathrm{supp}(\mu)$ and any $h$, by the construction of $u$. Since $x \in \mathrm{supp}(P[X])$, it follows that $\Pr(M_{j^\star} = m \,|\, X_{1:T} = x) = 0$ for all $m \in \mathrm{supp}(\mu)$, so $\mu^\top P[M_{j^\star} | X_{1:T} = x] = 0$.

We note that the statement about $\overline{G}_i(\widehat{e}(x))$ follows because of Lemma D.7. $\square$

*Proof of Lemma D.13.* To construct the value function, we define $\Theta^{(V)}$ in the same manner as Lemma D.4, such that $\Theta^{(V)}$ contains $\Theta^{(2,s)}$ constructed in Claim D.10 as a submatrix: $\Theta^{(V)}_{(m,j^\star,s),:} = \Theta^{(2,s)}_{m,:}$ for $s \in \mathcal{S}^\star$. All other rows of $\Theta^{(V)}$ are $\mathbf{0}$. It now follows that for $i \in \widehat{\mathcal{I}}$ and $x$ where $\Pr(\widehat{X}_{-i} = \widehat{x}_{-i}) > 0$, by definition of $\widehat{\mathcal{I}}$,

$$(\Theta^{(V)} \overline{G}_i(\widehat{e}(x))) \odot \phi(e(x_i)) = P[\widehat{X}_i = \widehat{x}_i, M_{J_i}, (J_i, S_i) | \widehat{X}_{-i} = \widehat{x}_{-i}]$$

The proof that this claim is correct follows the same reasoning as Lemma D.4, where we argue that $P[H_i \,|\, \widehat{X}_{-i} = \widehat{x}_{-i}]$ must concentrate on $\{j^\star\} \times \mathcal{S}^\star$ for all $i \in \widehat{\mathcal{I}}$. Thus, we can define $b = B^\top \mu$, where $B$ is defined in Lemma D.4. We observe that for $i \in \widehat{\mathcal{I}}$, the same reasoning as before gives

$$V(\overline{G}_i(\widehat{e}(x)), \widehat{e}_i(x)) = \mu^\top P[\widehat{X}_i = \widehat{x}_i, M_{j^\star} \,|\, \widehat{X}_{-i} = \widehat{x}_{-i}]$$

First, if $(\widetilde{z}, x) \notin \mathrm{supp}(P[\widehat{X}])$, by Claim D.15, we have $\mu^\top P[M_{j^\star} \,|\, X_{1:T} = x] = 0$. The expression above must also equal 0, as $(\widetilde{z}, x) \notin \mathrm{supp}(P[\widehat{X}])$. Otherwise, we have

$$V(\overline{G}_i(\widehat{e}(x)), \widehat{e}_i(x)) = \mu^\top P[M_{j^\star} \,|\, \widehat{X} = \widehat{x}] \Pr(\widehat{X}_i = \widehat{x}_i \,|\, \widehat{X}_{-i} = \widehat{x}_{-i})$$

Now we apply Claim D.14 to get the desired result in this case. A additional case is when $\Pr(\widehat{X}_{-i} = \widehat{x}_{-i}) = 0$. In this case, Claim D.15 shows that $\overline{G}_i(\widehat{e}(x)) = \mathbf{0}$, so it follows that the value function also computes 0 in this case.

Finally, we need to check the case where $\widehat{x} \notin \operatorname{supp}(P[\widehat{X}])$, and we want to show $V(\overline{G}_i(\widehat{e}(x)), \widehat{e}_i(x)) = 0$ for all $i > 1$. The case where $\Pr(\widehat{X}_{-i} = \widehat{x}_{-i}) = 0$ is already handled above. In the case where $\Pr(\widehat{X}_{-i} = \widehat{x}_{-i}) > 0$, we can apply Claim D.10 to our construction for $\Theta^{(V)}$ to get

$$
(\Theta^{(V)}\overline{G}_i(\widehat{e}(x)))_{m,h} = \begin{cases} P[M_{j^\star} = m, H_i = (j^\star, s) \mid \widehat{X}_{-i} = \widehat{x}_{-i}] & \text{if } h = (j^\star, s) \text{ for } s \in \mathcal{S}^\star \\ 0 & \text{otherwise} \end{cases}
$$

Thus, taking the element-wise product with $\phi(e(x_i)) = P[\widehat{X}_i = \widehat{x}_i \mid M_{J_i}, J_i, S_i]$, we must have, by Proposition D.1,

$$
((\Theta^{(V)}\overline{G}_i(\widehat{e}(x))) \odot \phi(e(x_i)))_{m,h} =
$$
$$
\begin{cases} P[\widehat{X}_i = \widehat{x}_i, M_{j^\star} = m, H_i = (j^\star, s) \mid \widehat{X}_{-i} = \widehat{x}_{-i}] & \text{if } h = (j^\star, s) \text{ for } s \in \mathcal{S}^\star \\ 0 & \text{otherwise} \end{cases}
$$

Both of these terms must be 0 since $\widehat{x} \notin \operatorname{supp}(P[\widehat{X}])$, giving the desired result. $\square$

Now we are ready to prove Theorem D.6.

*Proof of Theorem D.6.* The first case we consider is when $x \in \mathcal{Z}$, defined in (D.7). By applying Lemmas D.9 and D.13, we constructed key, query, and value functions for the attention head such that when $\widehat{\mathcal{I}}$ (D.8) is nonempty, the attended-to positions $\mathcal{I}$ satisfy $\mathcal{I} = \widehat{\mathcal{I}}$. In addition, by applying Lemma D.13, we also obtain that for $x \in \operatorname{supp}(P[X])$, $V(\overline{G}_i(\widehat{e}(x)), \widehat{e}_i(x)) = r_{x,i}\mu^\top P[M_{j^\star} \mid X_{1:T} = x]$. As the attention head averages $V(\overline{G}_i(\widehat{e}(x)), \widehat{e}_i(x))$ over the attended-to positions, and $r_{x,i}$ is positive for all $i \in \widehat{\mathcal{I}}$, we obtain the desired result.

In the second case, $x \notin \mathcal{Z}$, so $(\widetilde{z}, x) \notin \operatorname{supp}(P[\widehat{X}])$. By Lemma D.13, for all $i > 1$, the value function outputs 0. However, by the construction in Lemma D.9, the attention will only attend to $i > 1$. Thus, the output of the attention head is 0. However, Claim D.15 also implies that $\mu^\top P[M_{j^\star} \mid X_{1:T} = x] = 0$, giving the desired result. $\square$

# E   Experimental details

**Generating HMM parameters.**   For all experiments, we randomly generated the parameters of an HMM with 10 output symbols in its vocabulary. We generate a random transition matrix by taking a random convex combination of random permutation matrices. We mix as many permutation matrices as there are hidden states; i.e. if there are 4 hidden states, then we mix 4 random permutation matrices. The mixing weights are generated by sampling logits IID from a uniform distribution on $[0, 1]$ and then taking a softmax with temperature 0.01. Although this is a small temperature, the transition probabilities can still be around 0.7 for some transitions. The start distribution is also sampled in the same way, but with softmax temperature 10.0. The rows of the emission probability matrix is also sampled the same way with temperature 0.01.

**Pretrain model.**   The pretrained model follows the BERT-base architecture, except with 6 layers and a much smaller vocab size.

**Pretrain data and task.**   The pretraining data consists of 5000 sequences (documents) generated from the HMM, each with length 10240. We pretrain on this data by doing 5% masked LM on chunks of length 512. Pretraining runs for 3 epochs and takes about 5 hours on a single NVIDIA Tesla K80 GPU on 16-bit precision. We use an internal cluster for all experiments. Pretraining uses batch size 8 and learning rate 1e-5 with a linear warmup of 500 steps and linear decay schedule after 500 steps.

We generated 20 pretraining (and downstream) datasets for each problem instance and average over the 20 runs in the vanilla HMM comparison, while the memory-based distributions are run for 5 trials of pretraining and finetuning.

**Downstream.**   The downstream task samples a sparse ground truth linear weight $\mu$ with 6 nonzero elements. Positions for nonzero entries are sampled uniformly at random and values are sampled i.i.d. from a standard normal distribution. Although we do binary classification, we sample $\mu$ with 2 rows and take the label to be the argmax of the two scores, instead of having 1 row and taking the sign. We find that this results in less degenerate datasets (datasets where all labels are the same).

We generate 5000 training, 500 validation and 1000 test examples for the downstream tasks. Downstream training uses learning rate 0.01 for both prompt tuning and head tuning, with a linear warmup/decay schedule, for 5 epochs over the downstream data. We take the model returned at the last checkpoint as the result (no early stopping). We found that it was important to train prompt tuning with full precision, since the gradients are relatively small and become zero with discretization.

We used message passing in the HMM to compute the posterior distributions of the latent variables analytically.

**Prompt tuning.**   We prepended a length 20 continuous prompt to each sequence of input word embeddings. We initialize elements of the prompt vectors IID from the uniform distribution on $[-0.5, 0.5]$. Our implementation for prompt tuning used the code of [19], available at `https://github.com/kipgparker/soft-prompt-tuning`.