# OpenReview forum: "Why Do Pretrained Language Models Help in Downstream Tasks? An Analysis of Head and Prompt Tuning"
_NeurIPS.cc/2021/Conference — NeurIPS 2021 Spotlight_

### Official Review · Reviewer_Dxbb · 2021-07-14

**Rating:** 6
**Confidence:** 2

**Summary:**

This paper proposes an analysis framework to link the pre-training and downstream tasks. In this framework, the downstream task needs to predict the properties of the posterior distribution over latent variables in an underlying generative model. And When the generative model is a standard HMM, the downstream recovery is possible with a simple classification head under strong non-degeneracy assumptions. It further shows that changing the generative model to a memory-augmented HMM or using prompt tuning can relax the non-degeneracy conditions.

**Limitations And Societal Impact:**



**Main Review:**

Strengths:
This paper provides a theoretical analysis towards the relationship between the pre-training and downstream tasks. It proves that under strong non-degeneracy conditions on token emission probabilities, a linear layer can recover the labels in the downstream task. Also, it further proves recovery guarantees with relaxed non-degeneracy assumptions by using continuous prompt tuning, which explains the effectiveness of prompt tuning.

Weaknesses:
The data used in the experimental part are all synthetic and the pre-trained model is also trained based on synthetic data. How well can this theoretical analysis be fitted into the real data? How can the application side benefits from this theoretical analysis?

**Time Spent Reviewing:**

5 hours

---

> ### Author Response · Authors · 2021-08-10
> **Response**
>
> We thank the reviewer for the insightful review and questions. Responses are below.
>
> --- “Weaknesses: The data used in the experimental part are all synthetic and the pre-trained model is also trained based on synthetic data. How well can this theoretical analysis be fitted into the real data? How can the application side benefits from this theoretical analysis?”
>
> We would like to respectfully point out that the goal of our paper is, first and foremost, to introduce formal settings and mathematical abstractions to make a first-cut theoretical analysis possible. With this goal in mind, the experiments were also designed to verify key points from our theory.
>
> That being said, there are several directions for which we hope that our work can currently, or eventually, provide empirical insights. First, prompt-tuning is a very effective empirical method, and understanding how and why it works is a timely question for driving further practical improvements. Our work offers insight into how and why prompt tuning can be effective, as our analysis suggests that prompt tuning serves to condition the model to emit only task-relevant information and discard irrelevant information.
>
> Second, another direction for which we hope our style of work can eventually inform practice is in designing principled classification head architectures for adapting the model to the downstream task. As an example, our theory provides insights for when linear classification heads will or will not work on data from HMMs, which we later verify in Figure 3 (left). Section 4 also designs an attention head architecture which is provably effective for solving the downstream task when data is generated from a memory HMM, and Figure 3 (right) also verifies the efficacy of this architecture in simulation. Future work could perhaps help design other classification head architectures for the downstream task.
>
> Finally, regarding the applicability of our results to real data -- real data is complex and messy, and so simplifications such as ours are often *necessary* to obtain mathematical insights into the problem. We introduce the simplifying assumption that the data is generated by HMMs or memory HMMs. As these generative models can serve as simple but plausible models for real data [1], we believe it reasonable to hope that mathematical insights from analyzing HMMs or memory HMMs can eventually motivate improved algorithms for real data.
>
> References:
>
> [1] Justin T. Chiu and Alexander M. Rush. Scaling Hidden Markov Language Models. EMNLP 2020.

---

### Official Review · Reviewer_9BBf · 2021-07-16

**Rating:** 7
**Confidence:** 3

**Summary:**

The paper studies the question why head and prompt tuning work for pre-trained LM. Analyzing complicated neural networks is a daunting task. The paper offers an analysis framework based on the Hidden Markov Model (HMM) and memory augmented HMM. Under this framework, the paper makes a connection with masked LM (BERT) that at time step i, the output of BERT(`[mask]_i`) represents the marginal P(X_i) of the HMM. The framework further allows for analysis of a binary classification task using head tuning or prompt tuning via the posterior distribution of a HMM. To this end, the paper runs a simulation study with data generated from HMM and memory augmented HMM.

**Limitations And Societal Impact:**

The authors have adequately addressed the limitations and potential negative societal impact of their work.

**Main Review:**

- While pretrained models have been used intensively in NLP, much of the analysis on pretrained models is often carried out without considering down-stream tasks. This paper provides one of the first analyses on why pretrained LM does well in fine-tuning and prompt tuning for downstream tasks via a lens of generative models (HMM). While it poses some simplified assumptions (i.e., data generated from HMM), I think it is a valuable first step in this direction.

- The paper studies both head tuning and prompt tuning for both vanilla HMM and a more complex one, namely memory augmented HMM. Under each model, it shows that the downstream label can be recovered (i.e., predicted) using a linear layer on the conditional probability vector of the head.  While the analysis relied on some strict assumptions ( |H| << |V|), I think the paper presented quite a thorough analysis within its scope.

- Given that the paper focuses on MLM, which currently dominates the field of NLP, I think the simulation could be a bit more realistic. For example, one can train a large neural HMM [1] to generate language data. A MLM can be trained to verify if it can predict the marginal p(x_i) at position i.


**Time Spent Reviewing:**

4

---

> ### Author Response · Authors · 2021-08-10
> **Response**
>
> We thank the reviewer for the positive reviews and for noting that our work “provides one of the first analyses on why pretrained LM does well … for downstream tasks” and “is a valuable first step in this direction”. Responses to specific points are below.
>
> --- “ the analysis relied on some strict assumptions ( |H| << |V|)”
>
> To clarify, the non-degeneracy condition that $|H| <= |V|$ was only required for Theorem 3.1 regarding the vanilla HMM without prompt tuning. Subsequent sections study settings where the non-degeneracy conditions can be significantly relaxed, and, in particular, allow for the case that $|H| >> |V|$.
>
> --- “Given that the paper focuses on MLM, which currently dominates the field of NLP, I think the simulation could be a bit more realistic. For example, one can train a large neural HMM [1] to generate language data. A MLM can be trained to verify if it can predict the marginal p(x_i) at position i.”
>
> We thank the reviewer for the suggestion. To clarify, our simulations do train a BERT-style MLM on data generated from HMMs in order to verify key implications of our theory.
>
> The simulations performed BERT pretraining on synthetic HMMs due to the computational constraints; even then, we note that there was an increasing computational budget requirement to solve the MLM task well as the hidden state space grew. Computational budget allowing, the reviewer’s suggestion of training on semi-synthetic HMMs (with large hidden state size for modeling natural language) would be very interesting to incorporate in the next revision.
>
> The reviewer’s suggestion also alluded to the question of evaluating the model’s accuracy in predicting the conditional probabilities $P(X_i | x_{-i})$. To answer this, the model obtains the following tuples of (HMM hidden state size, average l2 error) on a held out set: [(4, 0.0381), (8, 0.163), (10, 0.188), (15, 0.21), (25, 0.228), (30, 0.226)]. With a vocabulary size of 10, this translates to an average error for a single coordinate of $0.226/\sqrt{10} = 0.0715$ when the hidden state size is 30. We note that the prediction of the conditional probabilities worsens with larger hidden state size as the language modeling task becomes more challenging and computational requirements increase.

---

### Official Review · Reviewer_xxSu · 2021-07-18

**Rating:** 7
**Confidence:** 2

**Summary:**

This work puts forward a theoretical analysis of pre-trained language models and head and prompt tuning on downstream tasks. In particular it studies the assumptions on the distribution of language and downstream tasks under which the downstream label can be recovered from a pretrained LM by using head or prompt tuning, by considering a generative model of language that can be represented as a HMM. The key components of the analysis include:
1. The data distribution is generated from a HMM (or a memory augmented HMM).
2. The initial state of the HMM, $H_0$, contains all the information necessary to recover the downstream label for the sequence.

Under this setup, for head tuning, the downstream labels are recoverable under the condition that the token emission probability matrix $W$ has linearly independent columns (implying that the dimensionality of the HMM hidden states is smaller than the vocabulary size of the data generating distribution). On the other hand, prompt tuning requires only a relaxed version of this condition, only the columns corresponding to the support of the downstream task need to be independent, eliminating the condition on the hidden state dimensionality.

Under memory augmented HMMs, for downstream tasks that can be represented as a linear function of the memory (with a fixed number of cells, $N$), the non-degeneracy condition can be relaxed even for head tuning. For prompt tuning, the independence assumption can be further relaxed to hold only for the set of memory cells that the downstream task depends on.

The authors further perform simulation experiments to validate the findings on different settings of the data generating distribution. Simulation results agree with the findings:
1. Prompt tuning outperforms head tuning in the setting where hidden state size is significantly larger than the token vocabulary size.
2. When the data is generated using memory-augmented HMMs, and the downstream task information can be recovered entirely from the memory, even head tuning gets close to perfect performance for all hidden state sizes.

**Main Review:**

Strengths:
1. Novel approach to analyze the effectiveness of pre-trained models on downstream tasks by modeling the generative distribution of language with a HMM.
2. Theoretically validates why the ongoing work on prompt tuning might be empirically more successful than head tuning.

Limitations:
1. The significance of the degeneracy conditions on the data-generating distribution and downstream tasks is not clearly discussed at points. Adding a short discussion on their practical implications could go a long way towards improving the impact of the work.

Other minor questions / comments:
1. Are there any significant differences in the distributions that can be modeled by a memory-augmented HMM as against a HMM, or is it fair to say that this is more of a limitation on the types of downstream tasks that can be recovered (those that can be expressed as a function of only the memory in the case of memory-augmented HMMs)?
2. Line 254: grasp -> grass

**Time Spent Reviewing:**

3.5

---

> ### Author Response · Authors · 2021-08-10
> **Response**
>
> We thank the reviewer for the positive review and insightful questions. Responses are below.
>
> --- “The significance of the degeneracy conditions on the data-generating distribution and downstream tasks is not clearly discussed at points. Adding a short discussion on their practical implications could go a long way towards improving the impact of the work.”
>
> We thank the reviewer for the suggestion and will incorporate the discussion below into the next revision.
>
> In the real world, the basic non-degeneracy condition in Assumption 3.1 is likely unrealistic because HMMs require a large state space to generate natural language [1]. Thus, relaxing the non-degeneracy condition is an important requirement for making the theory more realistic. This is accomplished by our results for prompt tuning and the memory HMM generative model. Our theoretical observations are in agreement with and can provide insight into the empirical effectiveness of prompt tuning.
>
> Empirically, insights from the non-degeneracy conditions potentially could also be applied to improve algorithmic performance in future work. For example, Assumption 3.1 suggests that increasing the size of the predicted vocabulary during pre-training could lead to better downstream task performance because it reduces the degeneracy (recall that $|H| > |V|$ leads to degeneracy in the vanilla HMM setting). A potential empirical instantiation of this idea is to increase vocabulary size by predicting n-grams during pre-training instead of single tokens.
>
> --- “Are there any significant differences in the distributions that can be modeled by a memory-augmented HMM as against a HMM, or [is this] more of a limitation on the types of downstream tasks that can be recovered … ?”
>
> Memory-augmented HMMs can be viewed as HMMs with a more structured transition function, so any distribution that can be modeled by a memory-augmented HMM can also be modeled by a vanilla HMM (with a large number of hidden states). Thus, it is accurate to view the memory HMM setting as a way to impose additional structure on the generative distribution and types of downstream tasks. The additional structure makes downstream recovery possible even when the hidden state space is large.
> A potential disadvantage of additional structure is that it often comes in hand with less realism. Fortunately, the memory HMM obtains the favorable side of this tradeoff, as it remains a plausible model for natural language (see Example 4.1) while allowing substantial improvements in downstream recovery..
>
> References:
>
> [1] Justin T. Chiu and Alexander M. Rush. Scaling Hidden Markov Language Models. EMNLP 2020.

---

### Official Review · Reviewer_33Yr · 2021-08-01

**Rating:** 7
**Confidence:** 3

**Summary:**

[UPDATE: Thanks to the authors for engaging with the comments in the review.  I hope to see them fully addressed in the revision and look forward to seeing the next version of the paper!]

The paper investigates how pretrained representations might be expected to help prediction on a downstream task.

In some sense, they should "obviously" help: the downstream predictor can learn to exploit any mutual information between its target output and the pretrained representations.  That's the standard motivation for multitask learning.

However, the paper exhibits specific settings in which the target output can be *exactly* predicted from the pretrained representations, using only a *simple* probe such as a linear classifier.

The paper is primarily theoretical, although a few experiments are included by way of illustration.  It is not clear what the experiments are intended to verify.

To make the analysis possible, the pretrained representations in the paper are not underlying representations such as word embeddings.  Rather, they are identifiable given the pretraining data, which makes it clear what "successful pretraining" means.  For example, the representation of word x_i in a sentence is the conditional distribution of x_i given all other words in the sentence, as predicted by a pretrained cloze language model.  The analysis assumes (though the experiments do not check) that the predicted conditional distribution actually matches the true conditional distribution.  This should be the case in the limit of infinite training data, provided that the cloze language model has adequate capacity and can be trained to a global optimum of log-loss.

It is shown that if the vocabulary is large enough, or under other assumptions, then this representation of x_1 is enough to recover the posterior distributions over h_1 and h_0, which are assumed to be sufficient for the downstream task.

Caveat: In the analysis, the observed sentences are assumed to be generated by a latent-variable model.  One would then expect the downstream task target output to be predictable from the actual latent variables.  However, the paper assumes more strongly that the target is a function of the *posterior distribution* of the latent variables given the observed sentences.  In other words, the goal of the downstream task is not to recover what the speaker *actually* meant, but only to recover what a hearer who knows the true generating distribution *thinks* it meant (given only the sentence and no other context).

Two structures are considered for the latent-variable model and its relationship to the downstream task.  For the more complex version, the predictor is also more complex, requiring argmax attention over all positions i.

In both cases, the paper considers allowing the predictor to prepend an arbitrary word embedding x_1 to the input.  This soft prompt can serve to shift the posterior predictions of x_i so as to be more informative for the downstream task.


**Main Review:**

I appreciate the style of work, as it goes beyond the usual "close your eyes and hope for the best" approach to neural networks.  It thinks carefully about what's going on with particular architectures.

It may be that the paper is answering a question that no one has asked.  As I noted in the summary, it seems obvious that pretrained features should help; in fact, it's known that even random features help!

However, I do think that this paper has some helpful insights.  Primarily about what can be extracted with a linear classifier for a downstream task, and why learned prompts and attention might also be useful.  The artificial setup is well chosen to make the pedagogical point.

The paper is precisely written, but was a slow read.  Although I'm very familiar in principle with all these topics, it was sometimes hard to connect the presentation to my usual concerns.  Since the primary goal of the paper is to aid understanding, I fear that the difficulty in skimming it will limit its impact.

I'm concerned that the graphs in the experiments may be comparing apples and oranges.  Are the points on each curve related at all, or is each point representing a single random HMM (with the error region showing different datasets generated from that HMM)?  If the 10-state HMM has nothing to do with the 15-state HMM, they probably shouldn't be on the same graph as if they're being compared -- the particular HMMs you chose may be unrepresentative.  You want to show mean and variance over many 10-state HMMs.  Similarly, is the right graph comparing some random memory HMM with some random vanilla HMM?  Again, that's just anecdotal, not systematic.  Also I'm not sure it's an interesting comparison since the tasks are different; I would have expected you to fix the setup to a memory HMM with its corresponding downstream task, and see what benefit was given by the attention architecture vs. the look-at-X_1 architecture from section 3.

---------------------------

SUGGESTIONS

To increase citations, I suggest including more high-level discussion of what is and isn't being shown.  I also suggest drawing more careful connections between the artificial settings in this paper and familiar real-world pretraining settings.  This hand-holding should probably be at the level of the comments I wrote in the paper summary.

The hardest thing to get my head around was the fact that you are (unusually) looking for exact recovery.  Just before section 3.1, I kept wondering why degeneracy was a problem.  After all, Bayes' Theorem still applies: it should *help* to be able to predict (in effect) X_0 even if X_0 doesn't have enough information to recover H_0 perfectly.  But for some reason, you want to be able to perfectly recover H_0 from *only* the prediction of X_0, which requires that prediction to preserve all of the relevant information from X_1, ..., X_n.

You are particularly interested in using a linear classifier for the downstream task.  Why this restriction?  You could say that the downstream task often has relatively little training data.  But that is a motivation for a low-parameter classifier, not a linear one.  In fact, you should admit that your classifier parameter \mu is high-dimensional if the vocabulary is large.

You rely heavily on the linearity of HMM inference.  You might want to defend why HMMs are plausible distributions for natural language.  The argument is presumably that with a large enough state space, it is possible to summarize an entire string prefix into a state.  A couple of papers relevant to this are https://arxiv.org/abs/2011.04640 and https://arxiv.org/pdf/1804.10747.pdf.  The paragraph at line 255 is also helpful.

Can you say anything about factorial HMMs?  The memory HMMs are only a simple case of factorial HMMs (2 factors, one being constant).

A little more hand-holding around proposition 3.4 would be helpful.  Your point is that predicting X_0 from X_{1:T} helps you predict H0 from X_{1:T}.  The reason for the complicated setup here is that X_0 is never actually observed (in fact you left it out of the model), but there are some stationarity invariants in the data -- and thus hopefully in the pretrained model -- that allow you to predict it anyway.

When using a prompt u as X_1, I think you want observing u to change the true posterior and hence the predicted posterior over X_2 (mediated by latent variables H_1 and H_2).  It would probably be helpful to say that explicitly.

Your construction with prompts assumes that the pretrained word embeddings are actually rows of the true emission matrix.  But how does that fit with the BERT pretraining that you do in your experiments?  Also, even if BERT pretraining works on real sentences, will it continue to get the "correct" cloze predictions when you prepend an embedding corresponding to a word never seen in training data?

I'm not clear on what the experiments are showing.  What do they add that was not present in the theoretical analysis?  I might have expected you to verify that BERT is in fact computing the correct word posteriors, and that the hidden state posteriors can be reconstructed from the word posteriors, so that you can predict downstream results for *any* mu, not just a particular random mu.

The assumption 3.5 is quite strong: it basically ensures that reconstructing one
   word is enough for the downstream task.  Maybe ok, but should be discussed more in
   the context of actual tasks, e.g., trying to get out some info and not other info.  (This assumption is relaxed in section 4.)

It might also be helpful to point out more explicitly that the cloze language model doesn't necessarily have to have the same structure as the true generating distribution.  Indeed in the experiments, it doesn't.  In the experiments, the data are generated by an HMM, but the cloze language model adopts a BERT architecture rather than doing something like spectral learning of the HMM.  It is implicitly assumed that BERT will be able to match the true posterior of the HMM, but this is not evaluated.

Regarding the caveat in my summary above: At line 138, it's a bit odd to think that the ground truth should be a linear function of the *posterior distribution* over H_0.  Ordinarily we would think that it would be determined by the *true value* of H_0, or at least, that it has high mutual information with H_0. However, if \mu_h = p(Answer=1 | H_0=h_0) - 0.5, then (3.1) is a good decision rule because it says to return 1 iff p(Answer=1 | X=x) >= 0.5, or equivalently, to return argmax_i p(Answer=i | X=x).  So perhaps you can argue that the ground truth is being labeled by a human who is using this decision rule to make their best guess given X=x, without privileged access to H_0.

---------------------------

MINOR REMARKS

The name "head tuning" seems misleading to me as it makes me think of attention heads in a Transformer, which is not the intention here.

On soft prompts, other work has recently appeared, including https://arxiv.org/abs/2104.06599, https://arxiv.org/pdf/2104.08691.pdf, https://arxiv.org/abs/2012.15723.

In section 4.1, the use of argmax with the attention head is nonstandard (section 4.1), although it's equivalent in the limit of scaling up key magnitude.  Should point this out.
The computation of the value functions is also nonstandard.

At line 114, I think you want to say more specifically that G(x) = \bar{G}(u,e(x)).
You say something like that later at 178.

At line 120, probably clearer to define P[U|V] first and then say that P[U|V=v] and P[U=u|V] select columns and rows from that matrix.

Line 188: shouldn't u be \tilde{z} here?

At line 283, by "article," do you mean words like "the" and "every"?
If so, I suggest using the word "determiner," which is more common in linguistics and NLP;
at first I thought you were talking about a news article, or perhaps just a
physical object (as in "article of clothing").

In Fig 3 (left), the "prompt tuning" line tunes the head AND the prompt, right?  Clear in the text but the figure legend and caption seem to say otherwise.

line 358: "ground-truth linear weight": do you mean that it's randomly generated just like the params at line 350?  If so, please use similar language in both places.  (What distributions are used?  I don't think you say, even in the appendices.)

**Time Spent Reviewing:**

6

---

> ### Author Response · Authors · 2021-08-10
> **Response [1/2]**
>
> We thank the reviewer for the very detailed and helpful feedback and for noting that the paper has “helpful insights … about what can be extracted with a linear classifier for a downstream task, and why learned prompts and attention might also be useful”. We will incorporate suggestions and address minor issues in the revision.
>
> --- “may be that the paper is answering a question that no one has asked … it seems obvious that pretrained features should help”
>
> While it’s perhaps “obvious” that pretrained features *should* help, the bigger question that our paper tackles is to understand *why* and *how* pretrained features help. E.g., as the reviewer also noted in a later sentence, it’s perhaps surprising that a simple linear probe can help solve the downstream task in certain settings. Moreover, we also show that prompt tuning can address more situations than a simple linear head.
>
> This is also an important question that a lot of people have asked; see for example the empirical work on probing information stored in BERT representations [1, 3, and other references in related work], which is motivated by understanding why BERT representations transfer so effectively to the downstream task.
>
> Our main contribution is the first theoretical analysis towards answering this question. Other related theoretical work [2] assumes the pretrained features help and explores implications but does not rigorously analyze why this assumption may hold. Towards analyzing why and how pretrained features help, an important first step is to introduce mathematical abstractions for the setting, as pretraining and finetuning deep networks is extremely challenging to analyze without further simplifications. We hope that follow-up work can build upon and improve the mathematical abstractions and first-cut analysis presented in this paper.
>
> Responses to questions about theoretical setup and contributions:
>
> --- “Caveat: ... [expected] ... target output to be predictable from the actual latent variables … paper assumes ... target is a function of the posterior distribution of the latent variables”
>
> We assume the target is a function of the posterior distribution over the latent variables because the posterior (not the exact latents) is the only information we can recover using only the sentence text. The reviewer also noted a similar point. For example, if the posterior distribution over the latents is very spread-out, we can’t hope to recover much about the actual latent variables. If the posteriors concentrate on single values of the latent variables, then our results can be directly translated into approximate recovery guarantees for functions of the actual latent variables. We omit this for a simpler analysis and presentation.
>
> --- “you are (unusually) looking for exact recovery. Just before section 3.1, I kept wondering why degeneracy was a problem … you want to be able to perfectly recover H_0”
>
> To clarify, degeneracy is an issue even if the goal is approximate, not exact, recovery. Consider the setting described in lines 281-290, where the posterior distribution of the syntax could concentrate on “determiner” (i.e. article POS) for certain positions. For such positions i, $P(X_i | H_i = h_i)$ would lie in a low rank subspace supported only on determiners, leading to a violation of Assumption 3.1 (non-degeneracy). As a result, $P(X_i | x_{-i})$ would not contain any semantic information, so the vanilla HMM recovery methods in Section 3 would not help with any recovery (even approximate) for a semantics-based downstream task. In this case, degeneracy is the fundamental reason why approximate recovery is not possible.
>
> --- “You are particularly interested in using a linear classifier for the downstream task. Why this restriction? … downstream task often has relatively little training data ... motivation for a low-parameter classifier”
>
> As noted by the reviewer, a low-parameter classifier is desirable for situations with relatively little training data, as is often the case for the downstream tasks.  We are particularly interested in linear heads because they are the simplest form of low-parameter classifiers. Though the parameter count of the linear head does increase with vocabulary size as noted by the reviewer, the linear head still requires fewer parameters than fine-tuning, and training the linear head remains a simple convex optimization problem. The simplicity of the linear head makes it an appealing candidate for studying basic possibilities and limitations for our problem setting.
>
> We also note that Section 4 studies a simple attention head, rather than linear head, designed for solving the downstream task. We envision that future work could study additional settings where more complicated classification heads, e.g. multilayer neural nets, are necessary for recovering downstream task labels.
>
> --- “... rely heavily on the linearity of HMM inference ... defend why HMMs are plausible distributions for natural language.”
>
> We thank the reviewer for suggesting the additional citations, which provide empirical support for using HMMs as a model for natural language. We will incorporate these citations into the revision. We agree with the reviewer’s point that HMMs are a plausible model for natural language because “with a large enough state space, it is possible to summarize an entire string prefix into a state.” HMMs also make a good starting point for the analysis because they are one of the most well-known generative models for language.
>
> However, as discussed in lines 162-164, the large state space of HMMs would lead to degeneracy, which would hamper recovery for the downstream task. One of our contributions is to analyze the memory HMM, which is more structured but still provides a plausible model for language. We exploit the additional structure in the memory HMM to relax the non-degeneracy assumptions for recovering ground-truth downstream labels.
>
> --- “Can you say anything about factorial HMMs? The memory HMMs are only a simple case of factorial HMMs (2 factors, one being constant).”
>
> We thank the reviewer for suggesting factorial HMMs -- this is certainly an interesting direction for future work that would require new analysis techniques and recovery methods. The analysis of Section 4 would not immediately apply to factorial HMMs because it requires the memory component to be constant.
>
> --- “construction with prompts assumes that the pretrained word embeddings are actually rows of the true emission matrix. But how does that fit with the … experiments?”
>
> Answering this question could be challenging because the model is largely a black box, so it is hard to determine how the model processes information from the embeddings. Consider the following possibilities, which would be hard to distinguish empirically: 1) word embeddings are obtained by transforming the rows of the true emission matrix into some alternative embedding space. The model internally inverts this (possibly complicated) transformation to obtain word emission probabilities, and proceeds to process these emission probabilities in the manner assumed by our theory, and 2) the model does something completely unrelated. Distinguishing 1) and 2) requires opening the black box of the model, which is challenging not only theoretically but also empirically.
>
> --- “will [BERT] … get the "correct" cloze predictions when you prepend an embedding corresponding to a word never seen in training data?”
>
> We thank the reviewer for this interesting question -- it would be hard to obtain a satisfactory answer because, if we’re understanding the question correctly, it seems to only apply when assumptions in Section 3.1 do hold. It is challenging to determine whether this is the case, as per our response to the question above.

---

> > ### Author Response · Authors · 2021-08-10
> > **Response [2/2]**
> >
> > Regarding the experiments:
> >
> > --- “not clear on what the experiments are showing … What do they add that was not present in the theoretical analysis?”
> >
> > The primary goal of the experiments is to verify key points of the theory in a setting where some of the assumptions do not exactly hold. For example, the assumption that the model $G$ outputs the conditional probabilities only holds approximately since we learn $G$ on data (see the next response). We will clarify this point in the revision.
> >
> > There are three important messages of the theory which the experiments aim to verify: 1) Degeneracy does inhibit downstream task recovery (Figure 3 left.) 2) Prompt-tuning helps with recovery in the setting where there is degeneracy (Figure 3 left.) 3) The additional structure imposed by the memory HMM does allow recovery with relaxed non-degeneracy conditions compared to the vanilla HMM (Figure 3 right.) We elaborate more on the comparison between memory HMM and vanilla HMM in a response below.
> >
> > --- “verify that BERT is in fact computing the correct word posteriors”
> >
> > On on a held out set, the tuples of (hidden set size, average l2 error in $P(X_1 | x_{-1})$) are as follows: [(4, 0.0381), (8, 0.163), (10, 0.188), (15, 0.21), (25, 0.228), (30, 0.226)]. Note that the vocabulary size is 10 throughout, so the average error for a single coordinate is $0.226/\sqrt{10} = 0.0715$ when the hidden state size is 30. The increase in l2 error with hidden state size is caused by the language modeling task becoming more challenging as hidden state size increases.
> >
> > ---  “concerned that the graphs in the experiments may be comparing apples and oranges. Are the points on each curve related at all[?] … You want to show mean and variance over many … HMMs”
> >
> > We thank the reviewer for raising this point -- due to computational constraints prior to the submissions deadline, we could only perform pre-training once per HMM, so the error bars in the current plots indeed only average over the downstream tasks. We have since rerun these experiments pretraining on 10 randomly generated HMMs per hidden state size and have found the same conclusions and trends to hold. The next revision will include these updated plots.
> >
> > --- “not sure [if memory HMM v.s. vanilla HMM is] an interesting comparison since the tasks are different”
> >
> > The comparison between the memory HMM and vanilla HMM in Figure 3 (right) verifies that downstream recovery does indeed improve when more structure is imposed on the data-generating distribution. Thus, we believe this empirical comparison is valuable because it agrees with one of our main theoretical results, which states that non-degeneracy conditions for downstream recovery can be substantially relaxed when data is drawn from a memory HMM v.s. vanilla HMM. We also note that, as per the response above, the trends in Figure 3 (right) still hold over multiple trials of pretraining on different HMMs.
> >
> > --- “[verify] that the hidden state posteriors can be reconstructed from the word posteriors, so that you can predict downstream results for any mu, not just a particular random mu.”
> >
> > It is actually crucial for our theory that the downstream result is only predicted for “a particular random mu” -- for the prompt tuning and memory HMM settings, it is unclear whether a single classification head can reconstruct the entire hidden state posterior. The strong non-degeneracy assumption (3.1) is required to guarantee reconstructing the entire hidden state posterior. For the settings where we can relax the non-degeneracy assumptions, our proofs rely on showing that the classification head can reconstruct information required for the particular downstream task determined by $\mu$ and ignore task-irrelevant information.
> >
> > Other clarifications:
> >
> > --- “the "prompt tuning" line tunes the head AND the prompt, right?”
> >
> > Yes.
> >
> > --- “ground-truth linear weight… What distributions are used?”
> >
> > We thank the reviewer for pointing out this omission. We generate the ground truth linear weight for the downstream task as a sparse matrix with k=6 non-zero entries (chosen to be non-zero uniformly at random). These non-zero entries are sampled IID from N(0, 1).
> >
> > References:
> >
> > [1] Anna Rogers, Olga Kovaleva, and Anna Rumshisky. A primer in bertology: What we know about how bert works. TACL 2020.
> >
> > [2] Nikunj Saunshi, Sadhika Malladi, and Sanjeev Arora. A mathematical exploration of why language models help solve downstream tasks. ICLR 2021.
> >
> > [3] Ian Tenney, Dipanjan Das, and Ellie Pavlick. Bert rediscovers the classical nlp pipeline. ACL 2019.

---

### Decision · Program_Chairs · 2021-09-27

**Decision:**

Accept (Spotlight)

**Comment:**

This paper provides a theoretical analysis of pretrained language models, linking the pretraining and downstream tasks with an underlying latent variable generative model of text. It analyzes head and prompt tuning in this setting, using a memory augmented HMM as the generative model in the analysis. The main findings are that under certain non-degeneracy conditions heads and prompt tuning can solve the downstream task with certain recovery guarantees. The theoretical findings are illustrated with experiments on synthetic data.

All reviewers agree that the theoretical analysis is novel and insightful, and a good first step in understanding the effectiveness of fine-tuning and prompt tuning. They pointed out as weaknesses lack of discussion on the practical implications, which the authors promised to address in the final version. While the use of synthetic data and lack of experimentation on more realistic tasks is a limitation, I believe the theoretical findings are useful and may foster future research considering more realistic scenarios.  I urge the authors to take into account the detailed comments made by the reviewers when preparing the final version.